# ENVIRONMENT PREDICTIVE CODING FOR VISUAL NAVIGATION

**Santhosh K. Ramakrishnan**[1,2]  **Tushar Nagarajan**[1,2]  **Ziad Al-Halah**[1]  **Kristen Grauman**[1,2]
[1]The University of Texas at Austin        [2]Facebook AI Research
{srama, tushar}@cs.utexas.edu,ziadlhlh@gmail.com,grauman@fb.com

## ABSTRACT

We introduce *environment predictive coding*, a self-supervised approach to learn environment-level representations for embodied agents. In contrast to prior work on self-supervised learning for individual images, we aim to encode a 3D environment using a series of images observed by an agent moving in it. We learn these representations via a *masked-zone prediction task*, which segments an agent's trajectory into zones and then predicts features of randomly masked zones, conditioned on the agent's camera poses. This explicit spatial conditioning encourages learning representations that capture the geometric and semantic regularities of 3D environments. We learn the representations on a collection of video walkthroughs and demonstrate successful transfer to multiple downstream navigation tasks. Our experiments on the real-world scanned 3D environments of Gibson and Matterport3D show that our method obtains 2 - 6× higher sample-efficiency and up to 57% higher performance over standard image-representation learning. Code and pre-trained models are publicly available: https://vision.cs.utexas.edu/projects/epc/

## 1 INTRODUCTION

In visual navigation, an intelligent embodied agent must move around a 3D environment using its stream of egocentric observations to sense objects and obstacles, typically without the benefit of a pre-computed map. Significant recent progress on this problem can be attributed to the availability of large-scale visually rich 3D datasets (Chang et al., 2017; Xia et al., 2018; Straub et al., 2019), and high-quality 3D simulators (Kolve et al., 2017; Savva et al., 2019a; Xia et al., 2020).

End-to-end reinforcement learning (RL) has been shown to achieve state-of-the-art navigation performance (Savva et al., 2019a; Wijmans et al., 2020). However, these approaches suffer from sample inefficiency and incur significant computational cost. Recent approaches try to mitigate these limitations by pre-training image representations offline and transferring them for navigation (Mirowski et al., 2016; Sax et al., 2020), or by performing auxiliary tasks and data augmentation in an online fashion during RL policy learning (Gordon et al., 2019; Kostrikov et al., 2021; Ye et al., 2020; 2021).

Current offline representation learning methods are flexible — once learned, the representations can be transferred to improve multiple embodied tasks. However, they are limited to learning image feature extractors (Gupta et al., 2017; Sax et al., 2020), or image-level proximity functions (Savinov et al., 2018; Chaplot et al., 2020c; Chang et al., 2020). Since embodied agents typically operate with limited field-of-view sensors, image representations only encode small parts of the scene in the nearby locality of the agent, and do not consider the broader context from the rest of the environment. We contend that embodied agents must learn higher-level semantic and geometric representations of the larger 3D environment around them, conditioned on their entire history of observations.

To that end, we introduce environment predictive coding (EPC), a self-supervised approach to learn *environment-level representations* that are transferrable to a variety of navigation-oriented tasks. The key idea is to learn an encoding of a 3D environment from a series of egocentric observations so as to be predictive of visual content that the agent has not yet observed. Consider the example in Fig. 1. An agent has observed the living room, lounge, and the bedroom in an unfamiliar house. The agent's encoding of the observed spaces (i.e., the red trajectory) should be predictive of the visual features at an unseen location $(x, y, \theta)$, e.g., the green viewpoint, and enable inferences like "there is a small kitchen", and "it contains a sink and an oven". Learning such predictive representations can equip an agent with the ability to reason about 3D environments as it starts performing various navigation-oriented tasks. The proposed EPC model aims to learn such representations that capture

Figure 1: **Environment Predictive Coding:** During self-supervised learning, our model is given video walkthroughs of various 3D environments. We mask out portions of the trajectory (dotted lines) and learn to infer them from the unmasked parts (in red). The resulting EPC encoder builds environment-level representations of the seen content that are predictive of the unseen content (marked with a "?"), conditioned on the camera poses. We then transfer this learned encoder to agents performing various navigation tasks in novel environments.

the natural statistics of real-world environments in a self-supervised fashion, by simply watching video walkthroughs recorded by other agents.

To achieve this, we devise a self-supervised *masked-zone prediction* task in which the model learns environment embeddings in an offline fashion, by watching pre-collected video walkthroughs recorded by other agents navigating in 3D environments (See Fig. 1). The videos contain RGB-D and odometry. Specifically, we segment each video, into zones of temporally contiguous frames which capture local regions of the 3D environment. Then, we randomly mask out zones, and predict features for the masked zones conditioned on both the unmasked zones' views and the masked zones' camera poses. Since the overlap in scene content across zones is typically limited, the model needs to reason about the geometry and semantics of the environment to figure out what is missing.

Our general strategy can be viewed as a context prediction task in sequential data (Devlin et al., 2018; Sun et al., 2019b; Han et al., 2019)—but, very differently, aimed at learning high-level semantic and geometric representations of 3D environments to aid embodied agents acting in them. Unlike any prior video-based feature learning, our approach learns features conditioned on camera poses, explicitly grounding them in a 3D space; we demonstrate the impact of this important distinction.

Through simulated experiments in photorealistic scenes from Matterport3D (Chang et al., 2017) and Gibson (Xia et al., 2018), we show that transferring the EPC environment-level representations leads to 2 - 6× higher sample-efficiency and up to 57% better performance compared to only image-level transfer on 4 navigational tasks: room goal navigation, object visitation, flee, and area coverage.

Our contributions are: (1) we propose environment-predictive coding (EPC), a self-supervised approach to represent the underlying 3D environment given the observation sequences of an embodied agent, (2) we propose the proxy task of masked-zone prediction to learn environment-level representations from video walkthroughs captured by other agents, (3) we perform extensive experiments on Gibson and Matterport3D to demonstrate that EPC leads to good improvements on multiple navigation-oriented tasks, and study EPC's design choices and noise robustness.

## 2 RELATED WORK

**Self-supervised visual representation learning**: Prior work leverages self-supervision to learn image and video representations from large unlabelled datasets. Image representation learning attempt proxy tasks such as inpainting (Pathak et al., 2016) and instance discrimination (Oord et al., 2018; Chen et al., 2020a; He et al., 2020), while video representation learning leverages signals such as temporal consistency (Jayaraman & Grauman, 2015; Wei et al., 2018; Kim et al., 2019) and contrastive predictions (Han et al., 2019; Sun et al., 2019a). VideoBERT (Sun et al., 2019a;b) jointly learns video and text representations from videos by filling in masked out information. Dense Predictive Coding (Han et al., 2019; 2020) learns video representations that capture the slow-moving semantics in videos. Whereas these methods tackle human activity recognition in videos, we aim to learn geometric and semantic cues in 3D spaces for embodied agents. Unlike the existing video models (Sun et al., 2019a;b; Han et al., 2019), which simply infer missing frame features conditioned on time, our approach explicitly grounds its predictions in the spatial structure of 3D environments.

**Representation learning via auxiliary tasks for RL**: Reinforcement learning approaches often suffer from high sample complexity, sparse rewards, and unstable training. Prior work tackles these using auxiliary tasks for learning *image-level* representations during online RL training (Mirowski

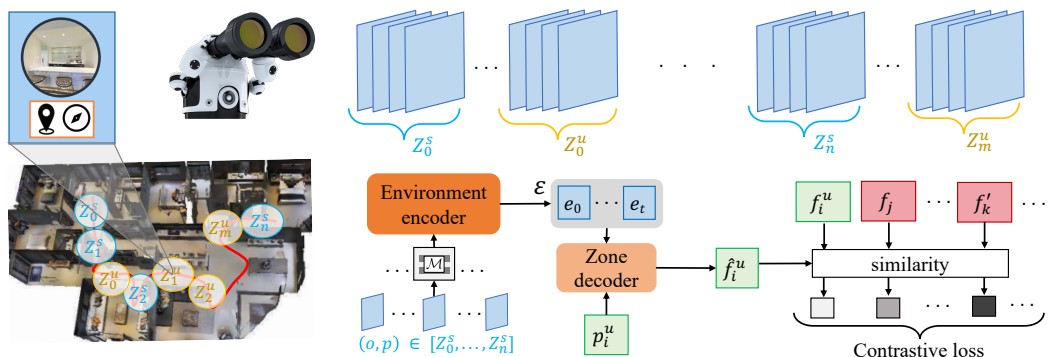

Figure 2: We propose the *masked-zone prediction task* for self-supervised learning of environment embeddings. We learn from video walkthroughs generated by other agents moving under policies ignorant of our eventual downstream tasks. Each frame consists of the egocentric view and camera pose (top left). We group the frames in the video into seen zones in cyan $\{Z_i^s\}_{i=0}^n$ and unseen zones in yellow $\{Z_i^u\}_{i=0}^m$ (top row). The zones are generated automatically by grouping temporally contiguous sets of frames (bottom left). Given a camera pose $p_i^u$ sampled from the unseen zone $Z_i^u$, we use a transformer-based encoder-decoder architecture that generates environment embeddings $\mathcal{E}$ from the seen zones, and predicts the feature encoding $\hat{f}_i^u$ conditioned on the pose $p_i^u$ (bottom center). The model is trained to distinguish the positive $f_i^u$ from negatives in the same video $\{f_j | f_j \neq f_i^u\}$ as well from other videos $\{f_k'\}$ (bottom right).

et al., 2016; Gordon et al., 2019; Shen et al., 2019; Ye et al., 2020; 2021; Stooke et al., 2021). In contrast, we learn *environment-level* representations in an offline fashion from previously captured videos. Recent work also learns state representations via future prediction and implicit models (Ha & Schmidhuber, 2018; Gregor et al., 2019; Hafner et al., 2019; Guo et al., 2020). In particular, neural rendering approaches achieve impressive reconstructions for arbitrary viewpoints (Eslami et al., 2018; Kumar et al., 2018). However, unlike our idea, they focus on pixelwise reconstruction, and their success has been limited to synthetically generated environments like DeepMind Lab (Beattie et al., 2016). In contrast to any of the above, we use egocentric videos to learn predictive feature encodings that capture the naturally occurring regularities of photorealistic 3D environments.

**Scene completion**: Prior scene completion work performs pixelwise (Jayaraman & Grauman, 2018; Ramakrishnan et al., 2019) or voxelwise (Song et al., 2017) reconstruction of 360 panoramas, image inpainting (Pathak et al., 2016), and image-level extrapolation of depth and semantics (Song et al., 2018; Yang et al., 2019). Pathdreamer furthers extends this to navigation in large-scale 3D scenes by predicting future observations, with the help of supervision from semantic annotations (Koh et al., 2021). Recent work on visual navigation extrapolates maps of room-types (Wu et al., 2019; Narasimhan et al., 2020) and occupancy (Ramakrishnan et al., 2020). While our approach is also motivated by anticipating unseen elements, we learn to extrapolate in a high-dimensional feature space (rather than pixels, voxels, or semantic categories) and in a self-supervised manner without relying on human annotations. Further, the proposed model learns from egocentric video sequences captured by other agents without assuming access to detailed 3D reconstructions.

**Learning representations for visual navigation**: Researchers are making steady advances in visual navigation architectures (Gupta et al., 2017; Chen et al., 2019; Fang et al., 2019; Wijmans et al., 2020; Chaplot et al., 2020b; Lenton et al., 2020). Prior work further improves navigation by pre-training representations using supervised image annotations (Gupta et al., 2017; Anderson et al., 2018b; Chen et al., 2019; Sax et al., 2020; Chaplot et al., 2020c), mined object relations (Yang et al., 2018), unannotated videos (Chang et al., 2020), and active exploration (Du et al., 2021). However, these approaches are limited to learning *image-level* functions. In contrast, we learn *environment-level* representations of image sequences conditioned on the camera poses. We show that the two are complementary by augmenting a SoTA navigation architecture from Fang et al. (2019).

## 3 APPROACH

We propose *environment predictive coding* (EPC) to learn environment-level representations via self-supervision on video walkthroughs (Sec. 3.1). To demonstrate the utility of these representations, we integrate them into a transformer-based architecture and refine them for individual navigation tasks (Sec. 3.2). Finally, we describe our procedure for generating video walkthroughs (Sec. 3.3).

### 3.1 ENVIRONMENT PREDICTIVE CODING

Our hypothesis is that it is valuable for an embodied agent to learn a predictive coding of the environment. The agent must not just encode the individual views it observes, but also learn to leverage the encoded information to anticipate the unseen parts of the environment. We propose to train an encoder-decoder model that observes a subset of views from a video walkthrough in a 3D environment, and then infers the features of unobserved views *conditioned on their camera poses*. To successfully infer features from unobserved views, the encoder must build a predictive representation of the underlying physical environment using the observed views. By transferring this encoder to a navigation agent, we equip the agent with the structural and semantic priors of 3D environments to quickly perform new tasks in new spaces, like mapping the house or room goal navigation.

We propose the self-supervised task of masked-zone prediction to achieve this goal (see Fig. 2). For this task, we use a dataset of egocentric video walkthroughs containing (possibly noisy) RGB-D and odometer sensor readings collected by other agents deployed in various unseen simulated environments (Fig. 2, top). These environments are inaccessible for interactive RL training, and the agent policies are ignorant of our eventual downstream tasks (see Sec. 3.3) Our method works as follows. First, we automatically segment each video into "zones" which contain temporally contiguous sets of frames. We then learn an environment encoder via the self-supervised masked-zone prediction task on the segmented videos. Finally, we transfer the learned environment encoder to an array of downstream navigation-oriented tasks. We explain each step in detail next.

**Zone generation** At a glance, one might first consider masking arbitrary individual frames in the training videos. However, doing so can result in poor representation learning since shared content from nearby unmasked frames can make the prediction task trivial. Instead, our approach masks *zones* of frames at once. We define a zone to be a set of temporally contiguous frames in the video. By choosing a large-enough temporal window, we can reduce the amount of shared content with temporally adjacent zones. Given a video walkthrough of size $L$, we divide it into zones $\{Z_0, Z_1, \cdots\}$ of length $N$ (selected through validation):

$$Z_i = \{(o_t, p_t) \mid \forall t \in [t_s, t_e]\}, \tag{1}$$

where $t_s = i \times N$, $t_e = \min((i+1) \times N, L)$, $o_t$ is the RGB-D sensor reading, and $p_t$ is the camera pose obtained by accumulating odometer readings from time $0$ to $t$ (see Fig. 2, bottom left). While two zones may share visual content, we find that this simple approach works better than strictly limiting the overlap between zones (see Appendix. A8). Thus, the learning is guided by predicting parts of the environment that were never seen as well as those seen from different viewpoints.

**Masked-zone prediction** Having segmented the video into zones, we next present our EPC masked-zone prediction task to learn environment embeddings (see Fig. 2). The main idea is to infer unseen zones in a video by previewing the context spanning multiple seen zones. We randomly divide the zones into seen zones $\{Z_i^s\}_{i=1}^n$ and unseen zones $\{Z_i^u\}_{i=1}^m$. Given the seen zones and the mean camera pose from an unseen zone $p_i^u$, we need to infer a feature encoding of the unseen zone $Z_i^u$. To perform this task efficiently, we first extract visual features $x_t$ from each RGB-D frame $o_t$ in the video using pretrained CNNs (described in Sec. 3.2). These features are concatenated with the corresponding pose $p_t$ and projected using an MLP $\mathcal{M}$ to obtain the image-level embedding. The target features for the unseen zone $Z_i^u$ are obtained by averaging[1] all the MLP projected features:

$$f_i^u = \frac{1}{|Z_i^u|} \sum_{\forall x \in Z_i^u} \mathcal{M}([x, \overrightarrow{0}]), \tag{2}$$

where we mask out the pose (i.e., $p = \overrightarrow{0}$) in the target to avoid trivial solutions. We use a transformer encoder-decoder model (Vaswani et al., 2017) to infer the zone features (see Fig. 2, bottom). An environment encoder uses self-attention over the image-level embeddings from all the seen zones, i.e., $\{\mathcal{M}([x, p]) \mid \forall (x, p) \in Z_i^s, \forall i \in [1, n]\}$, to generate the environment embeddings $\mathcal{E}$. A zone decoder then attends to $\mathcal{E}$ conditioned on the camera pose $p_i^u$ from the unseen zone and predicts the zone features:

$$\hat{f}_i^u = \text{ZoneDecoder}(\mathcal{E}, p_i^u). \tag{3}$$

Following Fang et al. (2019), we transform all poses in the input zones relative to $p_i^u$ before encoding, which provides the model an egocentric view of the world. We perform the same transformation

---

[1]We found this to be better than randomly sampling features within a zone (see Appendix A12).

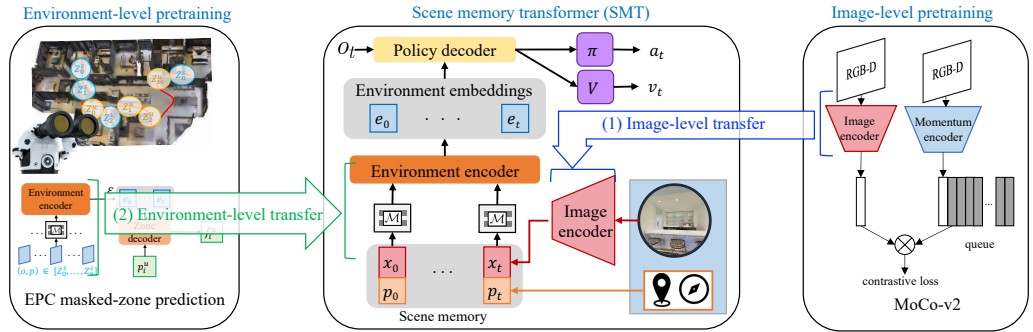

Figure 3: **Integrating environment-level pretraining for navigation. Right:** First, we transfer image-level representations to encode each RGB-D image. We transfer weights from an image encoder pre-trained for the self-supervised MoCo-v2 task on video walkthrough images. **Left:** Next, and most importantly, we transfer the environment-level representations to encode the entire history of observations. We transfer weights from the environment encoder and projection function $\mathcal{M}$ pre-trained for the proposed EPC masked-zone prediction task on video walkthroughs. **Center:** Finally, we finetune the SMT end-to-end on each task using RL.

to $p_i^u$ internally in the ZoneDecoder, effectively making it zero (i.e., no rotations or translations). As we will show in experiments, conditioning on pose is critical to learn useful representations. The environment encoder, zone decoder, and projection function $\mathcal{M}$ are trained end-to-end using noise-contrastive estimation (Gutmann & Hyvärinen, 2010). We use $\hat{f}_i^u$ as the anchor and $f_i^u$ from Eqn. 2 as the positive. We sample negatives from other unseen zones in the same video[2] and from all zones in other videos. The former discourages the model from simply learning video-specific textures and patterns; the latter expands the pool of negatives which was found to be beneficial in prior work (Chen et al., 2020b;a). The loss for the $i^{\text{th}}$ unseen zone in the video is:

$$\mathcal{L}_i = -\log \frac{\text{sim}(\hat{f}_i^u, f_i^u)}{\sum_j \text{sim}(\hat{f}_i^u, f_j) + \sum_k \text{sim}(\hat{f}_i^u, f_k')}, \tag{4}$$

where $f_k'$ are zone features from other videos, and $\text{sim}(q, k) = \exp\left(\frac{q \cdot k}{|q||k|} \frac{1}{\tau}\right)$ is a similarity measure with temperature $\tau = 0.1$. The idea is to predict zone representations that are closer to the ground truth, while being sufficiently different from the negative zones. Since the seen and unseen zones may only have limited overlap, the model needs to effectively reason about the geometric and semantic context in the seen zones to perform this task. We qualitatively analyse the masked-zone prediction results from the learned EPC model in Fig. 4.

## 3.2 Environment embeddings for embodied agents

Once a self-supervised environment encoder is trained, we transfer the encoder to various agents for performing navigation-oriented tasks. To this end, we integrate our pretrained environment encoder into the Scene Memory Transformer (SMT) from Fang et al. (2019). While our idea is potentially applicable to other memory models, our choice of SMT is motivated by the recent successes of transformers in NLP (Devlin et al., 2018) and vision (Sun et al., 2019b; Fang et al., 2019).

We briefly overview the SMT architecture (Fig. 3, center). We extract visual features from each RGB-D input (concatenated along the channel dimension) using a ResNet-18 image encoder (He et al., 2016). We store the visual features and agent poses $\{(x_i, p_i)\}_{i=0}^t$ observed during the episode in a scene memory. We then use an environment encoder to perform self-attention over the scene memory and generate a rich set of environment embeddings $\mathcal{E}$. We use a policy decoder to attend to $\mathcal{E}$ conditioned on the inputs $o_t = [x_t, p_t]$ at time t, and use the decoder outputs to sample an action $a_t$ and estimate the value $v_t$. We detail each component in the Appendix A2.

To incorporate our EPC environment embeddings into SMT, we first initialize the image encoder using CNNs pre-trained using a state-of-the-art MoCo-v2 method (Chen et al., 2020b) (see Fig. 3, right). The image encoder is pre-trained for 7,000 epochs on RGB-D images sampled from the video walkthroughs generated for EPC pre-training. Note that this is the CNN used to pre-extract RGB-D image features during EPC masked-zone prediction. Next, and most importantly, we initialize the environment encoder with our EPC pre-training on masked-zone prediction (see Fig. 3, left). We then finetune the SMT model end-to-end on the downstream task—whether that is room navigation,

---

[2]While the agent may revisit the same views in different zones, we find our learning to be robust to this.

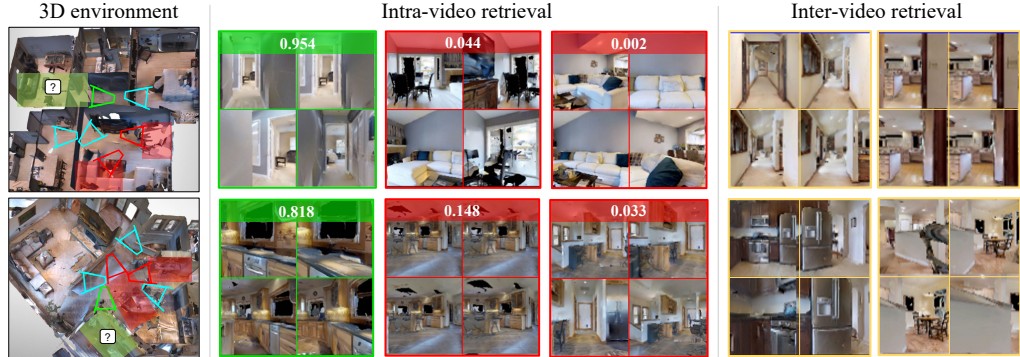

Figure 4: Each row shows one zone prediction example. **Left:** Top-down view of the 3D environment from which the video was sampled. Cyan viewing frusta correspond to the average pose for three input zones. Given the images and camera poses from each input zone, and a target camera pose (green frustum), the model predicts the corresponding zone feature (the masked green zone). **Center:** Given the inferred feature, we rank three masked (unobserved) zones from within the same video, where green is the positive zone and the red are the negatives. For each zone, we show four randomly sampled images along with the retrieval confidence. Our method retrieves the positive with high confidence. The model correctly predicts the existence of the narrow corridor (top row) and kitchen counter (bottom row) given the target poses. **Right:** Top two retrieved zones from *other* videos that are closest to the inferred feature. The predicted features are general enough to retrieve related concepts from other videos (here, narrow corridors and kitchens).

object visitation, flee, or area coverage (cf. Sec. 4). As we will demonstrate in the experiments, initializing the environment encoder using EPC leads to 2-6x higher sample efficiency and better performance when compared to initializing only the image encoder.

### 3.3 VIDEO WALKTHROUGH GENERATION

As discussed previously, EPC relies on a dataset of video walkthroughs containing RGB-D and odometer readings for self-supervision. Since EPC aims to learn the natural statistics of real-world environments (and not agent behavior), it suffices to use walkthroughs recorded in diverse environments by agents performing their day-to-day tasks. In particular, we do not require that the scenes used to generate walkthroughs are available for interactive policy learning, nor that the agent behaviors used for generating walkthroughs are tied to the downstream navigation tasks of interest. This means that we can, in parallel, collect videos from different agents operating in a large number of environments across geographical locations. We now realize this process in photorealistic Gibson scenes; we leave leveraging in-the-wild consumer videos as a challenge for future work.

We generate 2 sets of egocentric video walkthroughs by deploying simulated agents in photorealistic scanned indoor environments from Gibson[3] (Xia et al., 2018). **1) Strong Exploration:** an SMT agent that was trained for area coverage on MP3D, and **2) Weak Exploration:** a heuristic forward-biased navigation agent that moves forward until colliding, then turns randomly, and repeats. The weak exploration agent generates less-informative walkthroughs as it tends to repeatedly explore the same regions. Experimenting with EPC on both types of walkthrough allows us to test its dependence on walkthrough quality. Methods robust to this quality are desirable since it may be easier to collect lower quality walkthroughs on a large-scale. In both cases, the agents explore each Gibson environment starting from multiple locations for 500 steps per location and record the RGB-D and odometer readings. This results in 5,047 videos per agent, which we divide into an 80-20 train/val split (i.e., ∼2M training frames). As we will show in Sec. 4, the EPC encoders learned on these walkthroughs are applicable to navigation tasks not tied to the exploration agents.

## 4 EXPERIMENTS

First, we review the experimental setup (Sec. 4.1). We then evaluate the pre-trained EPC embeddings on multiple downstream navigation-oriented tasks in unmapped environments (Sec. 4.2). Finally, we present an ablation study of the data and proxy task used for EPC (Sec. 4.3).

---

[3]We use 332 Gibson environments from the Gibson 2+ training split (Wijmans et al., 2020).

### 4.1 EXPERIMENTAL SETUP FOR DOWNSTREAM TASKS

We perform experiments on the Habitat simulator (Savva et al., 2019b) with Matterport3D (MP3D) (Chang et al., 2017) and Gibson (Xia et al., 2018), two challenging and photorealistic 3D datasets with 90 and 572 scanned real-world indoor environments, respectively. Our observation space consists of $171 \times 128$ RGB-D images and agent pose $p = (x, y, \theta)$ relative to the starting pose at $t = 0$ (obtained by accumulating odometer readings). Our action space consists of: MOVE-FORWARD by 25cm, TURN-LEFT by $30°$, TURN-RIGHT by $30°$, and STOP (only for RoomNav). For all methods, we assume noise-free sensing during training, then evaluate with both noise-free and noisy sensing (pose, depth).

We perform interactive RL training on 61 MP3D train scenes. We evaluate the learned policies on 11 val and 18 test scenes in MP3D, and 14 val scenes in Gibson. These are disjoint from the 332 Gibson train scenes used for walkthroughs. We use episode lengths of $T = 1000$ for MP3D, and $T = 500$ for Gibson since the scenes are smaller. We further divide Gibson results into small and large environments (Ramakrishnan et al., 2020; Chaplot et al., 2020b). We evaluate our approach on four standard navigation tasks from the literature:
1. **Area coverage** (Chen et al., 2019; Chaplot et al., 2020b; Ramakrishnan et al., 2021): Agent is trained to maximize the area covered (in $m^2$) within a fixed time budget.
2. **Flee** (Gordon et al., 2019): Agent is trained to maximize the 'flee distance', i.e., the geodesic distance (in m) between its start and end positions, within a fixed time budget.
3. **Object visitation** (Fang et al., 2019; Ramakrishnan et al., 2021): Agent is trained to maximize the # object categories visited within a fixed time budget. An object is 'visited' if it is visible to the agent within 1.5m of the agent's position. We report both the # categories and # instances visited.
4. **RoomNav** (Savva et al., 2017; Wu et al., 2019; Narasimhan et al., 2020): Agent is trained to find the nearest room instance of a provided room category. We evaluate using the two standard metrics: SPL and Success (Anderson et al., 2018a). Please see Appendix A4 for the task details.

We choose these tasks since they capture different forms of geometric (tasks 1 & 2) and semantic (tasks 3 & 4) inference in 3D environments. All tasks are different from the agent behavior in the weak exploration trajectories. While task 1 is well-aligned with the strong exploration trajectories, all other tasks are distinct. Object visitation is a challenging task since the agent must navigate within 1.5m of each object and directly observe it. RoomNav is the most challenging task since the goal is to visit a specific room (not arbitrary exploration).

We compare our approach to several baselines.
**Scratch baselines**: We randomly initialize the visual encoders and policy, and train them end-to-end for each task. Images are encoded using ResNet-18. Agent pose and past actions are encoded using FC layers. *Reactive (scratch)* has no memory. *RNN (scratch)* uses a 2-layer LSTM as the temporal memory. *SMT (scratch)* uses a Scene Memory Transformer (SMT) for aggregating observations.
**SMT (MidLevel)**: This SMT-based model uses image encoders pre-trained for various supervised visual-tasks. The image encoders are kept frozen during RL (Sax et al., 2020; Zamir et al., 2020).
**SMT (MoCo)**: This SMT-based model uses a ResNet-18 image encoder pre-trained using MoCo-v2 (Chen et al., 2020b) and finetunes it end-to-end during RL. This is an ablation of our model from Sec. 3.2 where only the image encoder is pre-trained on the video walkthroughs. This SoTA image-level pre-training is critical to isolate the impact of our proposed environment-level pre-training.
**SMT (Video)**: This SMT-based model is inspired by Dense Predictive Coding (Han et al., 2019). The image encoder is initialized using weights from SMT (MoCo). It pre-trains the environment encoder as a 'video-level' model on the video walkthroughs. It takes 25 consecutive frames as input and predicts the features from the next 15 frames (following timespans used in Han et al. (2019)). During SSL, we mask the input camera poses and query based on time, unlike EPC which uses input poses and queries based on pose during SSL.
**ANS**: This is our implementation of the SoTA hierarchical policy from Active Neural SLAM (Chaplot et al., 2020b), upgraded to perform depth-based occupancy mapping (instead of using only RGB).

All RL agents receive RGB-D images and pose inputs. All models are trained for 13M-15M frames with 60 parallel processes. We train the scratch and ANS baselines for 2M more frames to account for the 2M frames in video walkthroughs used for other methods. Both SMT (MoCo) and SMT (Video) are given the advantage of using strong exploration walkthroughs for SSL (see Sec. 3.3). However, we test EPC with either strong or weak exploration walkthroughs (variants denoted as S.E and W.E, respectively). See Appendix A5 for optimization hyperparameters.

| Method | Area coverage (m$^2$) | | | Flee (m) | | | Object visitation (# i / # c) | | RoomNav (Succ. / SPL) | |
|---|---|---|---|---|---|---|---|---|---|---|
| | Gib-S | Gib-L | MP3D | Gib-S | Gib-L | MP3D | Gibson | MP3D | Gibson | MP3D |
| Reactive (scratch) | 28.2 | 50.1 | 121.8 | 3.0 | 4.0 | 6.6 | 6.4 / 4.0 | 12.1 / 4.9 | 1.5 / 0.4 | 0.9 / 0.2 |
| RNN (scratch) | 30.8 | 57.6 | 130.1 | 5.3 | 8.1 | 12.0 | 6.7 / 4.2 | 13.8 / 5.3 | 6.8 / 2.2 | 5.0 / 1.6 |
| SMT (scratch) | 32.2 | 62.6 | 166.5 | 4.3 | 6.4 | 11.0 | 9.2 / 5.4 | 17.8 / 6.3 | 5.0 / 2.3 | 3.0 / 1.4 |
| SMT (MidLevel) | 31.4 | 57.1 | 147.2 | 4.8 | 6.7 | 11.2 | 7.6 / 4.6 | 15.2 / 5.7 | 10.4 / 4.8 | 5.5 / 2.3 |
| SMT (MoCo) | 32.4 | 66.4 | 177.5 | 5.1 | 8.0 | 13.1 | 9.1 / 5.3 | 18.8 / 6.5 | 20.1 / 8.9 | 12.7 / 5.5 |
| SMT (Video) | 31.9 | 61.3 | 157.1 | 4.9 | 7.7 | 11.8 | 8.7 / 5.1 | 16.4 / 6.0 | 18.1 / 7.1 | 12.0 / 5.0 |
| ANS | 33.3 | **81.5** | 172.8 | 3.6 | 7.8 | 13.0 | 10.0 / 5.9 | 20.7 / 6.9 | 0.0 / 0.0 | 0.0 / 0.0 |
| EPC (W.E) | **33.9** | 76.1 | **195.1** | 5.9 | 10.6 | **16.6** | 11.9 / 6.5 | 24.0 / 7.6 | 28.9 / 12.6 | 21.1 / 9.7 |
| EPC (S.E) | **33.7** | 75.8 | **198.2** | 6.0 | 10.3 | 15.8 | 12.0 / 6.6 | 24.5 / 7.7 | 31.5 / 13.9 | 19.8 / 8.9 |

Table 1: **Downstream task performance** at the end of the episode. The two tasks in blue are geometric, and the two tasks in green are semantic. Gib-S/L means Gibson small/large. (# i / # c) means number of object instances/categories visited. All methods are trained and evaluated on 2 and 3 random seeds, respectively. In each column, the best methods are highlighted in bold (using a one-sided T-test with $p = 0.05$). We report only the mean due to space constraints. See Appendix A11 for performance vs. time step plots.

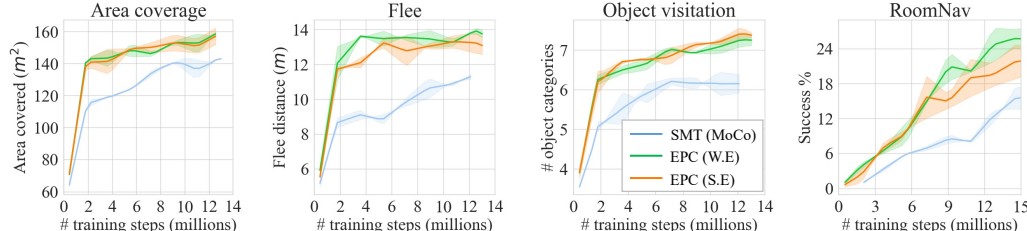

Figure 5: **Sample efficiency** on Matterport3D val split: We compare the sample-efficiency of EPC to that of the standard approach of pretraining only image-level features. Our environment-level pretraining leads to 2× to 6× better training sample efficiency on all four tasks. See Appendix A6 for corresponding Gibson plots.

## 4.2 DOWNSTREAM TASK PERFORMANCE

We transfer the EPC features to downstream navigation tasks. Tab. 1 shows the complete results. For brevity in the text, we also report the change in a method's performance relative to another as 4 values: (ΔAreaCoverage%, ΔFlee%, ΔObjectVisitation%, ΔRoomNav%), where each value is the mean relative change over all datasets and metrics for the corresponding task in Tab. 1.

SMT (scratch) outperforms RNN (scratch) and Reactive (scratch) on area coverage and object visitation. This is in line with results reported by Fang et al. (2019) and verifies our SMT implementation. Image-level pre-training from SMT (MoCo) provides good gains over SMT (scratch), particularly on Flee and RoomNav. It relatively improves over SMT (scratch) by (4%, 20%, 1%, 301%).

Environment Predictive Coding (EPC) effectively combines environment-level and image-level pre-training to substantially improve over only image-level pre-training in SMT (MoCo). EPC is most beneficial for the challenging semantic tasks of object visitation and RoomNav. For example, EPC (W.E) improves by (9%, 24%, 24%, 56%) relative to SMT (MoCo). Furthermore, SMT (Video)—the video-level pretraining strategy—significantly underperforms EPC, as it ignores the underlying spatial information during self-supervised learning. This highlights EPC's value in reasoning about the underlying 3D spaces of the walkthrough videos instead of treating them simply as video frames. We also don't observe a statistically significant difference between EPC (W.E) and EPC (S.E) on majority of the tasks. Thus, EPC is robust to the walkthrough quality, and does not require the agent behavior in walkthroughs to be tied to downstream tasks. This is advantageous since diverse and lower quality walkthroughs may be easier to obtain in practice.

Finally, EPC matches or outperforms the state-of-the-art ANS (Chaplot et al., 2020b) on both geometric and semantic tasks. We relatively improve over ANS by (2%, 42%, 13%, N/A) with EPC (W.E) and (2%, 40%, 15%, N/A) with EPC (S.E). On area coverage, while ANS performs better on Gibson, EPC outperforms it on the larger MP3D environments. On RoomNav, ANS fails to learn the STOP action resulting in 0% success. Note that ANS was not originally designed with a STOP action. Unlike the other agents with 4 actions, ANS has 24 × 24 location actions (see Appendix A3). Adding a STOP action to this huge action space makes RL optimization difficult.

In Fig. 5, we plot the validation metrics of SMT (MoCo) and EPC over the course of training. We see that environment-level pretraining offers significantly better sample efficiency: both EPC variants reach the best performance of SMT (MoCo) up to 6× faster. This confirms our hypothesis:

| Method | Area coverage (m$^2$) | | | Flee (m) | | | Object visitation (# i / # c) | | RoomNav (Succ. / SPL) | |
|---|---|---|---|---|---|---|---|---|---|---|
| | Gib-S | Gib-L | MP3D | Gib-S | Gib-L | MP3D | Gibson | MP3D | Gibson | MP3D |
| SMT (MoCo) | 32.4 | 66.4 | 177.5 | 5.1 | 8.0 | 13.1 | 9.1 / 5.3 | 18.8 / 6.5 | 20.1 / 8.9 | 12.7 / 5.5 |
| ANS | 33.3 | **81.5** | 172.8 | 3.6 | 7.8 | 13.0 | 10.0 / 5.9 | 20.7 / 6.9 | 0.0 / 0.0 | 0.0 / 0.0 |
| A. EPC | **33.9** | 76.1 | **195.1** | **5.9** | **10.6** | **16.6** | 11.9 / **6.5** | 24.0 / **7.6** | **28.9 / 12.6** | **21.1 / 9.7** |
| B. EPC w/o query-p | 32.2 | 63.0 | 166.1 | 5.0 | 7.0 | 11.4 | 7.8 / 4.8 | 15.3 / 5.8 | 12.0 / 4.9 | 8.1 / 3.3 |
| C. EPC w/ noisy-d | 33.4 | 72.5 | 185.6 | 5.6 | 8.6 | 15.0 | 12.2 / 6.7 | 24.0 / 7.6 | 27.3 / 11.6 | 19.0 / 8.0 |
| D. EPC w/ noisy-dp | 33.3 | 73.6 | **193.2** | 5.2 | 8.5 | 14.1 | 11.5 / 6.4 | 23.7 / 7.5 | 27.5 / 11.9 | 16.9 / 7.5 |
| E. EPC w/ noisy-d+vo | 33.6 | 74.7 | 192.1 | 5.8 | 9.8 | 15.5 | 11.6 / 6.4 | 23.3 / 7.5 | 25.3 / 10.5 | **19.2 / 8.3** |
| F. EPC w/o global | 33.1 | 72.0 | 187.2 | 5.0 | 7.9 | 12.9 | 11.2 / 6.3 | 22.7 / 7.4 | 28.1 / 12.4 | 20.4 / 9.2 |

Table 2: **Ablation study of EPC with weak exploration videos**. Top 2 rows are the best baselines from Tab. 1. Row B uses time to query the representation during masked-zone prediction (not pose). Rows C, D show the impact of sensory noise (depth and/or odometer) in the video walkthroughs. Row E shows the impact of replacing GT pose with estimates from a visual-odometry model (vo). Row F re-defines the masked-zone prediction task to use a local input context (instead of global). See Appendix A7 for ablations with EPC (S.E).

transferring environment-level encoders learned via spatial reasoning helps embodied agents learn faster compared to the current approach of transferring image-level encoders alone (Sax et al., 2020).

### 4.3 ABLATION STUDY

We now present an ablation study of EPC in Tab. 2 to answer three key questions.

**1. Does EPC need spatial conditioning for SSL?** One of our key contributions is to infer the unseen zone features conditioned on pose $p_i^u$ from the unseen zone (see Eqn. 2). We now try conditioning only on the time $t_i^u$ when the agent visited the unseen zone. Note that the dataset, inputs to the environment encoder, and the loss function still remain the same. Consider rows A and B in Tab. 2. The performance for EPC w/o query-p dramatically *declines* by (12%, 25%, 30%, 61%) relative to EPC. Thus, it is critical to use spatial queries for SSL with EPC.

**2. Does EPC need noise-free sensors to gather videos for SSL?** In Tab. 1, we assumed noise-free depth and odometer sensors for gathering video walkthroughs. We now inject noise into the depth and odometer readings from videos to analyze EPC's robustness. We use popular existing noise models: the depth noise uses disparity-based quantization, high-frequency noise, and low-frequency distortion (Choi et al., 2015); the odometry noise is based on data collected from a LoCoBot, and accumulates over time leading to drift in pose (Ramakrishnan et al., 2020; Chaplot et al., 2020b). Additionally, we replace the GT pose with pose estimates from RGB-D based visual odometry (Zhao et al., 2021). Consider rows A, C, D, and E in Tab. 2. EPC is robust to sensory noise in the videos and its transfer performance is stable. The performance declines on area coverage and flee when we inject depth noise (noisy-d), and on object visitation and RoomNav when we further inject odometry noise (noisy-dp). The performance remains similar when using estimated pose from visual odometry (noisy-d+vo). Overall, EPC retains its advantages over the best baselines from Tab. 1.

**3. Does EPC need global input context during SSL?** We use the global input context from multiple seen zones for EPC masked-zone prediction (Fig. 2). We now reduce this context to a more localized part of the video. Specifically, we modify the task to use the local context from a randomly sampled chunk of 25 contiguous frames as input, and infer features from the next 15 frames conditioned on pose. Consider rows A and E in Tab. 2. When we remove the global context (EPC w/o global), the performance deteriorates by (3%, 20%, 4%, 2%) relative to EPC. As shown in Appendix A7, we observe an even more drastic reduction of (7%, 17%, 8%, 21%) for EPC w/o global with the strong exploration videos. This suggests that the global input context is beneficial for SSL with EPC.

In Appendix A9, we demonstrate the robustness of EPC's policies to sensory noise during testing.

## 5 CONCLUSIONS

We introduced Environment Predictive Coding, a self-supervised approach to learn environment-level representations for embodied agents. By training on video walkthroughs generated by other agents, our model learns to infer missing content through a masked-zone prediction task. When transferred to multiple downstream embodied agent tasks, the resulting embeddings lead to 2-6× higher sample efficiency and upto 57% better performance when compared to the current practice of transferring only image-level representations. Our work highlights the advantages of self-supervised learning in agent videos by predicting features grounded in the spatial structure of 3D environments. In future work, we aim to extend our idea to in-the-wild videos and apply it to multimodal tasks such as embodied question answering, instruction following, and audio-visual navigation.

# 6 ACKNOWLEDGEMENTS

UT Austin is supported in part by the IFML NSF AI Institute, the FRL Cognitive Science Consortium, and DARPA Lifelong Learning Machines. We thank members of the UT Austin Vision Lab for feedback on the project. We thank Dhruv Batra for feedback on the paper draft. We thank the ICLR reviewers and meta-reviewers for their valuable feedback and suggestions.

# 7 REPRODUCIBILITY STATEMENT

We describe our method in Sec. 3 with architecture diagrams and equations to support the textual description. We describe our experimental setup in Sec. 4.1, and provide the necessary hyperparameters in Appendix A5. We publicly release the code and pre-trained models here: https://vision.cs.utexas.edu/projects/epc/.

# 8 ETHICS STATEMENT

Our idea for self-supervised learning relies on having a dataset of video walkthroughs captured in a diverse set of environments. In our experiments, we demonstrate a proof-of-concept by generating these walkthroughs in photorealistic Gibson scenes using simulated agents. However, obtaining such a large-scale dataset of walkthroughs in real-world environments can be challenging due to privacy concerns. To create such a dataset, we would need to carefully consider the privacy of the people and personal belongings recorded as a part of this data.

We perform experiments on the publicly available Gibson and Matterport3D datasets. Since these datasets primarily contain data from work places and economically well-off residences, our results may be influenced by the social and geographical distribution of the scanned buildings in these datasets. In experiments, we demonstrate good generalization of embodied policies trained in MP3D train scenes to novel MP3D test scenes and novel Gibson val scenes. However, it is possible that the generalization may be impacted when the policies are evaluated in environments that are significantly outside the distribution of these datasets.

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

# Appendices

We provide additional information about the experimental settings as well as quantitative results to support the experiments from the main paper. Below is a summary of the sections in the Appendix:

## A1  SUPPLEMENTARY VIDEO

The supplementary visualization "epc_visualization_supp.pdf" provides a brief overview of environment predictive coding and the masked-zone prediction task. We also show an example of inter-video retrievals (similar to Fig. 4 from main paper). In the demonstrated example, we show that our EPC model is able to accurately retrieve the masked-zone from the current scene, and even retrieve similar zones from other scenes.

## A2  SCENE MEMORY TRANSFORMER

We provide more details about individual components of the Scene Memory Transformer (Fang et al., 2019). As discussed in the main paper, the SMT model consists of a ResNet-18 visual encoder that extracts features $x_t$ from RGB-D inputs, and a scene memory for storing the visual features $\{x_i\}_{i=0}^t$ and agent poses $\{p_i\}_{i=0}^t$ seen during an episode. The environment encoder uses self-attention on the scene memory to generate a richer set of environment embeddings $\{e_i\}_{i=1}^t$. The policy decoder attends to the environment embeddings using the inputs $o_t$ at time t, which consist of the visual feature $x_t$, the agent pose $p_t$, and optionally, the navigation goal $g_t$. The outputs of the policy decoder are used to sample an action $a_t$ and estimate the value $v_t$. Next, we discuss the details of the individual components.

**VISUAL ENCODER**    At each time step $t$, the visual encoder extracts features $x_t$ from the RGB-D input.

$$x_t = \text{VisualEncoder}([r_t, d_t]) \tag{5}$$

where $r_t$ and $d_t$ are the RGB and depth inputs, respectively. We consider two visual encoders for our work. The first variant is a modified ResNet-18 encoder from (Wijmans et al., 2020) where the number of output channels are halved and the BatchNorm layers are replaced by GroupNorm. We used this encoder for the scratch models, SMT (MoCo), SMT (Video), and the EPC variants in the main paper. Next, we consider MidLevel features derived from various pre-trained CNNs to solve midlevel perception tasks (Sax et al., 2020; Zamir et al., 2020). For RGB inputs, we extract features from the pre-trained models in the max-coverage set proposed in Sax et al. (2020). These include 4 encoders trained for predicting surface normals, keypoints, semantic segmentation, and 2.5D segmentation. For depth inputs, we extract features from pre-trained models that predict

surface normals and keypoints from depth (Zamir et al., 2020). These encoders are kept frozen during reinforcement learning, following Sax et al. (2020). MidLevel features were used for SMT (MidLevel) in the main paper. We do not use the MidLevel encoder for EPC since we found the ResNet-18 encoder to be more efficient and achieved better performance.

**SCENE MEMORY** It stores the visual features $x_t$ derived from the input images and the agent poses $p_t$ at each time-step $t$. Following Fang et al. (2019), the pose $p_t$ consists of the 2D coordinates $(x, y)$ of the agent, heading $\theta$, and the time $t$. While we assumed that the ground-truth pose $(x_t, y_t, \theta_t)$ was available for downstream tasks in the main experiments, we evaluate the impact of noisy odometry in Appendix A9.

**ATTENTION MECHANISM** Following the notations from Vaswani et al. (2017), we define the attention mechanism used in the environment encoder and policy decoder. Given two inputs $X \in \mathbb{R}^{n_1 \times d_x}$ and $Y \in \mathbb{R}^{n_2 \times d_y}$, the attention mechanism attends to $Y$ using $X$ as follows:

$$\text{Attn}(X, Y) = \text{softmax}\left(\frac{Q_X K_Y^T}{\sqrt{d_k}}\right) V_Y \tag{6}$$

where $Q_X \in \mathbb{R}^{n_1 \times d_k}, K_Y \in \mathbb{R}^{n_2 \times d_k}, V_Y \in \mathbb{R}^{n_2 \times d_v}$ are the queries, keys, and values computed from $X$ and $Y$ as follows: $Q_X = XW^q$, $K_Y = YW^k$, and $V_Y = YW^v$. $W^q, W^k, W^v$ are learned weight matrices. The multi-headed version of Attn generates multiple sets of queries, keys, and values to obtain the attended context $C \in \mathbb{R}^{n_1 \times d_v}$.

$$\text{MHAttn}(X, Y) = \text{FC}([\text{Attn}^h(X, Y)]_{h=1}^H). \tag{7}$$

We use the transformer implementation from PyTorch (Paszke et al., 2019). Here, the multi-headed attention block builds on top of MHAttn by using residual connections, LayerNorm (LN) and fully connected (FC) layers to further encode the inputs.

$$\text{MHAttnBlock}(X, Y) = \text{LN}(\text{MLP}(H) + H) \tag{8}$$

where $H = \text{LN}(\text{MHAttn}(X, Y) + X)$, and MLP has 2 FC layers with ReLU activations.

We now describe the memory attention mechanisms used in SMT. At time $t$, the agent recieves features $x_t$, pose $p_t$, and optionally the navigation goal $g_t$. First, we transform the pose vectors $\{p_i\}_{i=1}^n$ from the scene memory relative to the current agent pose $p_t$. This allows the agent to maintain an egocentric view of the past inputs (Fang et al., 2019). Next, the environment encoder performs self-attention between the features stored in the scene memory $M$ to obtain the environment encoding:

$$\mathcal{E} = \text{EnvironmentEncoder}(M) = \text{MHAttnBlock}(M, M) \tag{9}$$

Next, the policy decoder attends to the environment encodings $\mathcal{E}$ using a linear projection of the current observation $f = \mathcal{M}([x_t, p_t, g_t])$.

$$\mathcal{D} = \text{PolicyDecoder}([x_t, p_t, g_t], \mathcal{E}) = \text{MHAttnBlock}(f, \mathcal{E}) \tag{10}$$

**POLICY** It consists of two FC layers corresponding to the actor $\pi$ and the critic $V$. Given the output $\mathcal{D}$ of the PolicyDecoder, the policy samples an action $a_t \sim \pi(\mathcal{D})$ and predicts the value function $v_t = V(\mathcal{D})$. Please see the next section for more details on the exact architectures used.

## A3 MODEL ARCHITECTURES

We provide the architectures for the RNN (scratch), SMT (scratch), and ANS baselines in Fig. A1. For Reactive (scratch), we remove the "Transformer Encoder" and "Transformer Decoder" blocks from SMT (scratch) and directly feed the input features to the policy. For SMT (MidLevel), we replace the ResNet encoder with the MidLevel encoders discussed in Sec. A2. For SMT (MoCo), SMT (Video), and EPC, we use the same architecture as SMT (scratch) and initialize pre-trained weights according to the method used.

For ANS, Fig. A1 describes a global policy that samples as actions an (x, y) location or STOP. The input occupancy map is a $300 \times 300 \times 2$ egocentric map where the two channels track occupied

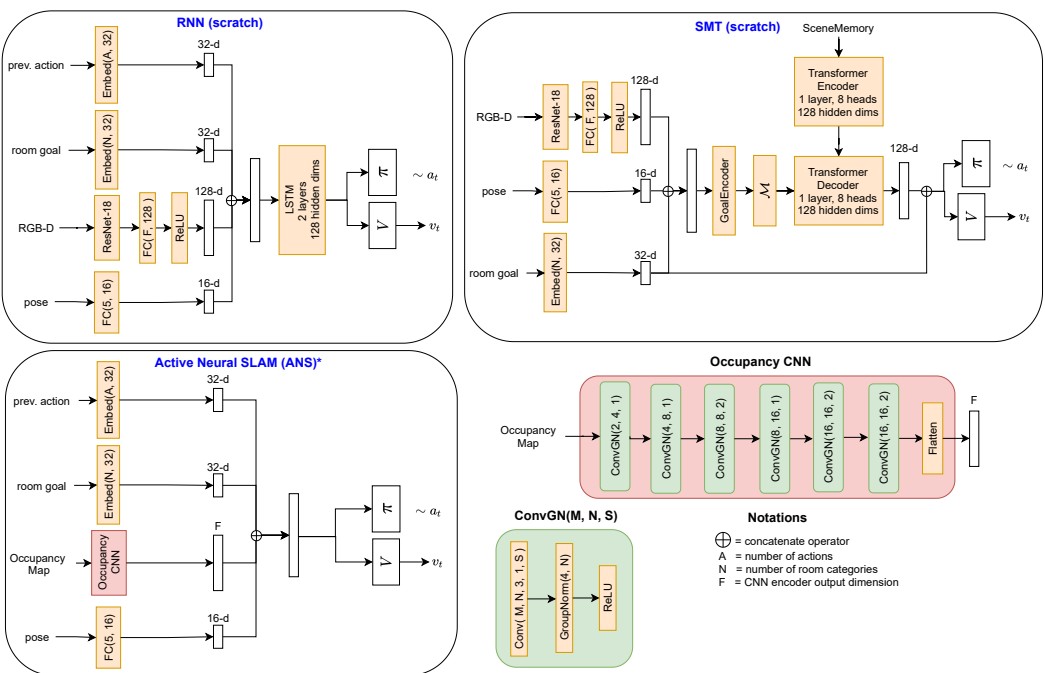

Figure A1: **Baseline architectures**: We show the detailed architectures for each of the key baselines. "GoalEncoder" is a 1-layer MLP. $\mathcal{M}$ is a 2-layer MLP. Embed, FC, Conv, GroupNorm, and ReLU are PyTorch layers corresponding to `nn.Embedding()`, `nn.Linear()`, `nn.Conv2d()`, `nn.GroupNorm()`, and `nn.ReLU()`, respectively (Paszke et al., 2019). The architectures shown are used for the RoomNav task. For the remaining tasks, we remove the room goal "Embed" and "GoalEncoder" layers and keep the rest of the architecture unchanged.

and explored regions. It captures a $48\text{m} \times 48\text{m}$ region around the agent. A local policy, which is a low-level navigator, then navigates to the (x, y) location for 25 steps. We use an analytical mapper+planner as the local policy as this was found to perform as well as a learned policy (Chaplot et al., 2020b;a). For area coverage, flee, and object visitation, we use a $48\text{m} \times 48\text{m}$ action space for the global policy. We try both discrete ($24 \times 24$ actions) and continuous action spaces, and pick the best choice based on the validation performance. For RoomNav, we use a discrete action space and add STOP as an additional action, giving us $24 \times 24 + 1 = 577$ actions.

## A4    ADDITIONAL TASK DETAILS

We provide more details about the object visitation and RoomNav tasks below.

**OBJECT VISITATION**    To determine if an object is visited, we check if it is within 1.5m of the agent, present in the agent's field of view, and if it is not occluded Ramakrishnan et al. (2021). We use a shaped reward function that rewards the agent for visiting a new object category and a cell-coverage reward to encourage exploration (similar to  Fang et al. (2019)):

$$R_t = O_t - O_{t-1} + 0.02(C_t - C_{t-1}), \tag{11}$$

where $O_t$, $C_t$ are the number of object categories and 2D grid-cells visited by time $t$. For MP3D, we use all 21 object categories defined for the ObjectNav task in Habitat (Batra et al., 2020; Savva et al., 2019a):

```
chair, table, picture, cabinet, cushion, sofa, bed, chest of drawers,
plant, sink, toilet, stool, towel, tv monitor, shower, bathtub, counter,
fireplace, gym equipment, seating, clothes
```

For Gibson, we use the semantic annotations from the 3D Scene Graph dataset (Armeni et al., 2019). We evaluate on the following categories:

```
chair, dining table, cook, vase, bottle, couch, bed, refrigerator, potted
plant, sink, toilet, clock, tv, oven, cup, umbrella, bowl, bench
```

We report both the number of categories and instances of the above objects visited. For MP3D, we evaluate on 550/900 validation/test episodes. For Gibson, we evaluate on 300 validation episodes. We do not split Gibson semantic results into small/large since there are only 6 val semantic scenes.

**ROOMNAV**   For an agent to successfully reach a specified room category, it needs to be within $0.1$m of a navigable location inside any room belonging to that category. We use a shaped reward function (Savva et al., 2019a):

$$R_t = d_{t-1} - d_t + 2.5S_t - 0.001 \tag{12}$$

where $d_t$ is the distance to the nearest target room location, $S_t$ indicates success at time $t$, and $0.001$ is a slack penalty. We evaluate on Gibson and MP3D rooms of the following categories:
```
bathroom, bedroom, office, kitchen, living room, dining room
```

We measure both success rate and SPL, i.e., success weighted by path length (Anderson et al., 2018a). We calculate SPL similar to (Batra et al., 2020), where the shortest path length is defined based on the target room closest to the agent's starting point. We evaluate on 410/1600 val/test episodes on MP3D, and 598 val episodes on Gibson. We use 500-step episodes following Narasimhan et al. (2020) (unlike 1000-step episodes for other tasks on MP3D).

## A5   HYPERPARAMETERS

All models are trained in PyTorch (Paszke et al., 2019) with DD-PPO (Wijmans et al., 2020) for 13M-15M frames with 60 parallel processes. Since SMT (MoCo), SMT (Video) and EPC benefit from 2M frames of off-policy experience in the video walkthroughs, we train the scratch and ANS baselines for 2M more frames to account for this. Note that SMT (MidLevel) is already pre-trained on 4M frames of annotated data.

We detail the list of hyperparameter choices for different tasks and models in Tab. A1. For ANS, we use 4 PPO mini-batches, 4 PPO epochs (not 2), and entropy coefficient of 0.003. For SMT (Video), we randomly sample 40 consecutive frames in the video and predict the final 15 frames from the initial 25 frames (following time-spans from Han et al. (2019)). For EPC , we randomly mask out $m$ zones of size $N$ in the video and predict them from the remaining video. We selected these values based on a grid-search and the validation performance on downstream tasks. For area coverage, flee, and object visitation, we found $N = 5, m = 6$ to work best. For RoomNav, we found $N = 5 - 40, m = 6$ to work best on weak exploration videos, and $N = 40, m = 6$ to work best on strong exploration videos. The hyperparameter search results are shown in Figs. A2, A3.

## A6   SAMPLE EFFICIENCY ON GIBSON

We plot the Gibson validation performance as a function of training experience in Fig. A4. EPC achieves 2 - 6× higher sample efficiency through environment-level pre-training when compared to the only image-level pre-training from SMT (MoCo).

## A7   ABLATIONS OF EPC WITH STRONG EXPLORATION VIDEOS

In Sec. 4.3 from the main paper, we presented an ablation study of EPC (W.E). We now provide the corresponding results for EPC (S.E) in Tab. A2. Our conclusions remain the same as in Sec. 4.3.
(1) EPC needs spatial conditioning, i.e., to query using pose during SSL (instead of only time).
(2) EPC is robust to noise in depth and odometer sensors used for collecting the video walkthroughs.
(3) EPC benefits from global input context during SSL (instead of a local input context of 25 frames).

## A8   COMPLEX ZONE CREATION SCHEMES FOR EPC

Over the course of arriving at our final EPC zone creation scheme, we tried an alternative scheme that attempted to strictly minimize the overlap between the seen and unseen zones. We measured

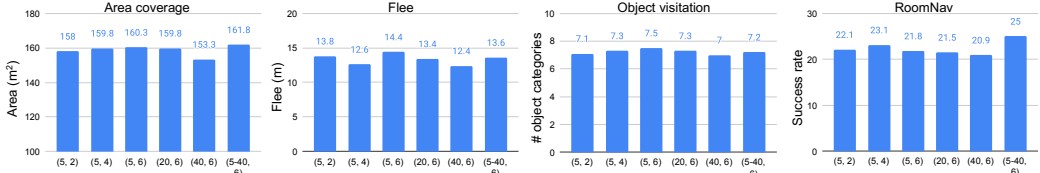

Figure A2: **EPC hyperparameter search on weak exploration videos:** We show the MP3D validation results for different zone sizes on 3 tasks. The X-axis indicates the zone size as $(N, m)$ where $N$ is the number of frames in each zone, and $m$ is the number of zones masked in the video. For $N = 5\text{-}40$, we randomly sample $N$ from 5 to 40 for each video during training.

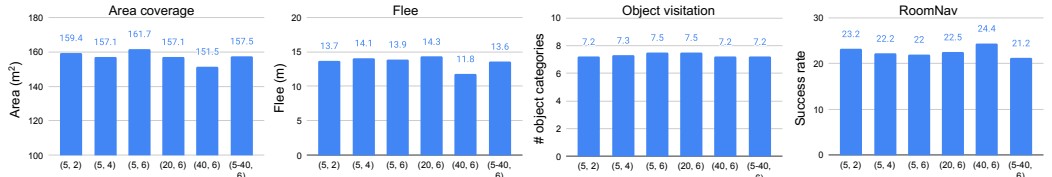

Figure A3: **EPC hyperparameter search on strong exploration videos:** We show the MP3D validation results for different zone sizes on 3 tasks. The X-axis indicates the zone size as $(N, m)$ where $N$ is the number of frames in each zone, and $m$ is the number of zones masked in the video. For $N = 5\text{-}40$, we randomly sample $N$ from 5 to 40 for each video during training.

| RL Optimization | |
| --- | --- |
| Optimizer | Adam |
| Learning rate | 0.00025 |
| # parallel actors | 60/64 |
| PPO mini-batches | 2* |
| PPO epochs | 2* |
| PPO clip param | 0.2 |
| Value loss coefficient | 0.5 |
| Entropy coefficient | 0.01 |
| Advantage estimation | GAE |
| Normalized advantage? | Yes* |
| Training episode length | 1000* |
| GRU history length | 128 |
| # training steps (in millions) | 13-15 |
| SMT hyperparameters | |
| Hidden size | 128 |
| Scene memory length | 500 |
| Scene memory stride | 2/4 |
| # attention heads | 8 |
| # encoder layers | 1 |
| # decoder layers | 1 |
| Self-supervised learning optimization | |
| Optimizer | Adam |
| Learning rate | 0.0001 |
| Video batch size | 10 |
| Temperature ($\tau$) | 0.1 |

Table A1: Hyperparameters for training our RL and self-supervised learning models. * - we use 4 PPO mini-batches and 4 PPO epochs for ANS. We disable normalized advantage and use 500-step episodes for RoomNav

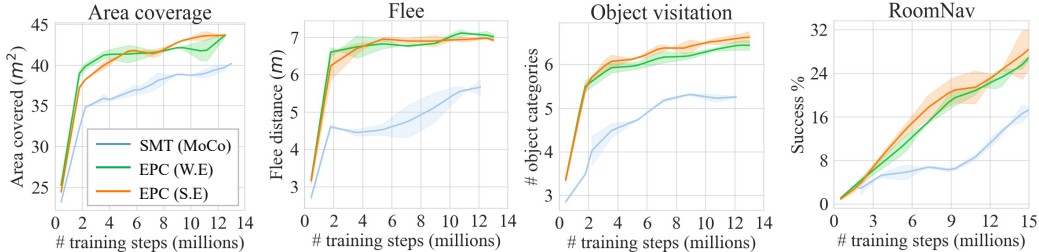

Figure A4: **Sample efficiency** on Gibson val split. Our environment-level pre-training leads to 2 - 6× higher sample efficiency when compared to SoTA image-level pre-training.

| | Area coverage (m²) | | | Flee (m) | | | Object visitation (# i / # c) | | RoomNav (Succ. / SPL) | |
|---|---|---|---|---|---|---|---|---|---|---|
| Method | Gib-S | Gib-L | MP3D | Gib-S | Gib-L | MP3D | Gibson | MP3D | Gibson | MP3D |
| SMT (MoCo) | 32.4 | 66.4 | 177.5 | 5.1 | 8.0 | 13.1 | 9.1 / 5.3 | 18.8 / 6.5 | 20.1 / 8.9 | 12.7 / 5.5 |
| ANS | 33.3 | **81.5** | 172.8 | 3.6 | 7.7 | 13.0 | 10.0 / 5.9 | 20.6 / 6.9 | 0.0 / 0.0 | 0.0 / 0.0 |
| A. EPC | **33.7** | 75.8 | **198.2** | **6.0** | **10.3** | **15.8** | **12.0 / 6.6** | **24.5 / 7.7** | **31.5 / 13.9** | **19.8 / 8.9** |
| B. EPC w/o query-p | 31.9 | 62.7 | 162.9 | 5.3 | 7.8 | 12.5 | 8.0 / 4.8 | 15.6 / 5.9 | 13.1 / 5.6 | 10.0 / 4.3 |
| C. EPC w/ noisy-d | **33.8** | 74.9 | **196.4** | 5.7 | **10.2** | **15.9** | **11.8 / 6.5** | **24.3 / 7.7** | 29.4 / **12.7** | 19.6 / 8.4 |
| D. EPC w/ noisy-dp | 33.6 | **75.8** | 192.2 | 5.1 | 8.3 | 13.7 | **12.0 / 6.7** | **24.1 / 7.8** | 26.6 / 11.7 | **17.7 / 8.2** |
| E. EPC w/o global | 32.4 | 68.4 | 178.4 | 5.1 | 8.1 | 13.2 | 10.8 / 6.0 | 22.1 / 7.2 | 23.7 / 9.6 | **17.4** / 7.4 |

Table A2: **Ablation study of EPC with strong exploration videos**. Top 2 rows are the best baselines from Tab. 1. Row B uses time to query the representation during masked-zone prediction (not pose). Rows C, D show the impact of sensory noise (depth and/or odometer) in the video walkthroughs. Row E defines the masked-zone prediction task to use a local input context (instead of global). All methods are trained and evaluated on 2 and 3 random seeds, respectively, and we report the mean. In each column, the best methods are highlighted in bold (using a one-sided T-test with $p = 0.05$).

the 3D point-cloud overlap between every pair of frames in a video walkthrough (projected using depth, camera pose, and camera intrinsics), and clustered the frames that have significant overlap together using hierarchical agglomerative clustering (Lukasová, 1979). We defined each cluster to be a zone, and used the same SSL training pipeline from Sec. 3. Tab. A3 shows the results on the MP3D validation split on 3 tasks. The spatial overlap variant works reasonably well. However, it is significantly more complex than our original proposal (requiring camera intrinsics, 3D point-cloud overlap computations and clustering), and performs worse. Since the videos are unconstrained, the agent might revisit the same areas multiple times, causing too many frames to be grouped together into a single zone. This results in very irregular zone sizes for masked-zone prediction across videos. This affects the quality of EPC SSL training. By not strictly limiting the overlap between zones, the agent learns by both inferring seen features (but from different viewpoints), as well as unseen features. Our current method is simpler, and works better.

## A9 NOISE ROBUSTNESS OF LEARNED POLICIES

In the main paper, we assumed the availability of ground-truth depth and pose sensors for downstream tasks (In Tab. 2, we added pose and depth noise to the walkthrough videos only). Now, we relax these assumptions and re-evaluate all methods by injecting noise in the depth and odometer sensors for downstream tasks (same noise models from prior work that we applied in Sec. 4.3), without any noise-correction. This is a common evaluation protocol for assessing noise robustness (Chen et al., 2019; Ramakrishnan et al., 2021; Chen et al., 2021). Note that the noise in agent's pose estimate increases over time since the per-step noise in odometer readings are accumulated. We compare the noise-robustness of all methods in Tab. A4. As expected, the performance declines as we add noise to more sensors (depth, then pose). We observe larger deterioration in performance when noise is added to depth. Note that this is likely due to the domain shift in the depth inputs between noise-free training and noisy evaluation. Nevertheless, most approaches are reasonably stable. EPC outperforms all methods when all noise sources are added. ANS declines rapidly in the absence of noise-correction due to accumulated map errors.

| Method | Flee (m) MP3D | Object visitation # categories | # instances | RoomNav (MP3D) SPL | Success |
|---|---|---|---|---|---|
| EPC (W.E) w/ spatial | 12.0 | 7.4 | **26.4** | 9.8 | 23.7 |
| EPC (W.E) w/ temporal | **14.4** | **7.5** | 26.3 | **11.0** | **25.0** |
| EPC (S.E) w/ spatial | 12.5 | 7.3 | 25.7 | 9.3 | 22.5 |
| EPC (S.E) w/ temporal | **13.9** | **7.5** | **26.6** | **10.3** | **24.4** |

Table A3: **Comparing zone creation schemes for EPC**. 'EPC w/ spatial' is a variant of EPC which uses 3D spatial overlap between frames to cluster them into zones. 'EPC w/ temporal' is our original method proposed in Sec. 3. The results are on the MP3D validation split. Our temporal scheme works better while being much simpler to implement.

## A10 ANS PERFORMANCE ON FLEE

ANS relies on a global policy that samples a spatial goal location for navigation. A local navigation policy then executes a series of low-level actions to reach that goal. Our qualitative analyses indicate that the global policy overfit to large MP3D environments. It often samples far away exploration targets, relying on the local navigator to explore the spaces along the sampled direction. However, this strategy fails in the small Gibson-S environments (typically a single room). Selecting far away targets results in the local navigator oscillating in place trying to exit a single-room environment. This does not affect area coverage much because it suffices to stand in the middle of a small room and look at all sides.

## A11 DOWNSTREAM TASK PERFORMANCE VS. TIME

We show the downstream task performance as a function of time in Fig. A5. For each model, we train with 2 random seeds and evaluate with 3 random seeds. We report the mean and the 95% confidence interval in the plots. EPC converges faster than using only image-level pretraining in SMT (MoCo), and outperforms a state-of-the-art Active Neural SLAM (ANS) on most tasks.

## A12 MASKED-ZONE TASK: AVERAGING VS. RANDOM SAMPLING FEATURES

In Eqn. 2 from the main paper, we averaged the projected visual features for all frames within a zone to get the ground-truth zone feature. Now, we compare an alternative strategy where we randomly sample a visual feature from a frame in the zone instead of averaging. Specifically,

$$f_i^u = M([x_i, \overrightarrow{0}]), \text{where } x_i \sim Z_i^u \tag{13}$$

We compare this with our original feature averaging strategy on the MP3D val set for flee, object visitation, and roomnav in Tab. A5. As we can see, our current scheme of averaging the projected features over all frames in the zone works better.

## A13 RELATION TO VISUAL PLACE RECOGNITION

Prior work on visual place recognition build maps and recognize whether the current visual information is from a place recorded in the map (Lowry et al., 2015). A place could be an exact location (Kuipers, 2000), rooms (Kuipers, 2000), or locally distinctive regions (Kuipers & Byun, 1991; Bailey et al., 1999). Zones in EPC can be viewed as a specific 'place' in 3D environments. However, instead of learning place-level features for recognizing previously seen places (Chen et al., 2017), we learn encoders by inferring features for unseen places.

| | Area covered (m$^2$) | | | | | | | | |
|---|---|---|---|---|---|---|---|---|---|
| | Gibson-S | | | Gibson-L | | | MP3D | | |
| Method | NF | N-D | N-D,P | NF | N-D | N-D,P | NF | N-D | N-D,P |
| Reactive (scratch) | 28.2 | 29.9 | 30.2 | 50.1 | 49.2 | 50.8 | 121.8 | 119.5 | 120.9 |
| RNN (scratch) | 30.8 | 33.0 | 33.1 | 57.6 | 56.4 | 56.9 | 130.1 | 106.6 | 108.8 |
| SMT (scratch) | 32.2 | 34.5 | 34.6 | 62.6 | 62.8 | 62.1 | 166.5 | 153.8 | 149.3 |
| SMT (MidLevel) | 31.4 | 33.2 | 33.0 | 57.1 | 54.5 | 53.7 | 147.2 | 143.0 | 138.3 |
| SMT (MoCo) | 32.4 | 34.5 | 35.1 | 66.4 | 66.4 | 66.2 | 177.5 | 152.5 | 148.9 |
| SMT (Video) | 31.9 | 34.3 | 33.7 | 61.3 | 61.9 | 59.4 | 157.1 | 138.5 | 137.4 |
| ANS | 33.3 | 35.6 | 33.3 | **81.5** | **84.4** | 62.7 | 172.8 | 171.7 | 129.6 |
| EPC (W.E) | **33.9** | **36.4** | **36.0** | 76.1 | 77.0 | **75.6** | 195.1 | 179.7 | 175.2 |
| EPC (S.E) | **33.7** | **36.5** | **36.4** | 75.8 | 77.1 | **75.7** | 198.2 | 183.2 | 178.9 |

| | Flee distance (m) | | | | | | | | |
|---|---|---|---|---|---|---|---|---|---|
| | Gibson-S | | | Gibson-L | | | MP3D | | |
| Method | NF | N-D | N-D,P | NF | N-D | N-D,P | NF | N-D | N-D,P |
| Reactive (scratch) | 3.0 | 3.0 | 3.1 | 4.0 | 4.1 | 4.1 | 6.6 | 7.8 | 7.9 |
| RNN (scratch) | 5.3 | 5.0 | 5.0 | 8.1 | 7.5 | 7.4 | 12.0 | 11.1 | 10.9 |
| SMT (scratch) | 4.3 | 4.2 | 4.2 | 6.4 | 6.5 | 6.5 | 11.0 | 10.8 | 10.9 |
| SMT (MidLevel) | 4.8 | 4.6 | 4.5 | 6.7 | 6.1 | 6.1 | 11.2 | 10.8 | 10.5 |
| SMT (MoCo) | 5.1 | 5.1 | 5.0 | 8.0 | 8.0 | 8.0 | 13.1 | 12.3 | 12.1 |
| SMT (Video) | 4.9 | 4.9 | 4.4 | 7.7 | 7.6 | 7.0 | 11.8 | 10.9 | 10.2 |
| ANS | 3.6 | 3.5 | 3.0 | 7.7 | 7.8 | 6.3 | 13.0 | 12.7 | 11.2 |
| EPC (W.E) | **5.9** | **5.9** | **6.0** | **10.6** | **10.6** | **10.5** | **16.6** | **15.7** | **15.1** |
| EPC (S.E) | **6.0** | **5.9** | 5.9 | **10.3** | **10.2** | 10.0 | 15.8 | **15.2** | **14.7** |

| | Object visitation (# instances /# categories) | | | | | |
|---|---|---|---|---|---|---|
| | Gibson | | | MP3D | | |
| Method | NF | N-D | N-D,P | NF | N-D | N-D,P |
| Reactive (scratch) | 6.4 / 4.0 | 6.2 / 3.8 | 6.3 / 3.9 | 12.1 / 4.9 | 11.5 / 4.7 | 11.6 / 4.8 |
| RNN (scratch) | 6.7 / 4.2 | 6.4 / 4.1 | 6.3 / 4.0 | 13.8 / 5.3 | 12.1 / 5.0 | 12.0 / 4.9 |
| SMT (scratch) | 9.2 / 5.4 | 8.8 / 5.1 | 8.6 / 5.1 | 17.8 / 6.3 | 16.7 / 6.1 | 16.1 / 6.0 |
| SMT (MidLevel) | 7.6 / 4.6 | 7.4 / 4.4 | 7.4 / 4.4 | 15.2 / 5.7 | 13.9 / 5.4 | 13.5 / 5.3 |
| SMT (MoCo) | 9.1 / 5.3 | 8.8 / 5.3 | 8.8 / 5.3 | 18.8 / 6.5 | 18.2 / 6.5 | 18.0 / 6.5 |
| SMT (Video) | 8.7 / 5.1 | 8.3 / 5.0 | 8.2 / 5.0 | 16.4 / 6.0 | 15.2 / 5.8 | 14.7 / 5.7 |
| ANS | 10.0 / 5.9 | 9.2 / 5.5 | 6.6 / 4.1 | 20.6 / 6.9 | 18.8 / 6.6 | 11.1 / 4.7 |
| EPC (W.E) | **11.9 / 6.5** | 11.3 / **6.3** | **11.2 / 6.3** | **24.0 / 7.6** | **23.2 / 7.6** | **22.2 / 7.4** |
| EPC (S.E) | **12.0 / 6.6** | **11.9 / 6.5** | **11.8 / 6.5** | **24.5 / 7.7** | **23.6 / 7.6** | **23.2 / 7.5** |

| | RoomNav (SPL / Success) | | | | | |
|---|---|---|---|---|---|---|
| | Gibson | | | MP3D | | |
| Method | NF | N-D | N-D,P | NF | N-D | N-D,P |
| Reactive (scratch) | 0.4 / 1.5 | 0.3 / 1.0 | 0.3 / 1.1 | 0.2 / 0.9 | 0.2 / 0.6 | 0.2 / 0.7 |
| RNN (scratch) | 2.2 / 6.8 | 2.3 / 6.2 | 2.1 / 5.6 | 1.6 / 5.0 | 1.5 / 4.8 | 1.5 / 4.8 |
| SMT (scratch) | 2.3 / 5.0 | 2.1 / 4.9 | 1.7 / 3.8 | 1.4 / 3.0 | 1.7 / 3.7 | 1.6 / 3.5 |
| SMT (MidLevel) | 4.8 / 10.4 | 4.3 / 10.5 | 4.5 / 9.9 | 2.3 / 5.5 | 1.9 / 4.4 | 1.7 / 3.9 |
| SMT (MoCo) | 8.9 / 20.1 | 7.8 / 18.1 | 7.7 / 17.4 | 5.5 / 12.7 | 4.9 / 11.0 | 4.0 / 8.7 |
| SMT (Video) | 7.1 / 18.1 | 6.1 / 16.3 | 4.8 / 12.5 | 5.0 / 12.0 | 4.3 / 10.1 | 3.6 / 8.4 |
| ANS | 0.0 / 0.0 | 0.0 / 0.0 | 0.0 / 0.0 | 0.0 / 0.0 | 0.0 / 0.0 | 0.0 / 0.0 |
| EPC (W.E) | **12.6 / 28.9** | **12.2 / 27.2** | **11.0 / 24.0** | **9.7 / 21.1** | **8.4 / 18.1** | **7.5 / 15.7** |
| EPC (S.E) | **13.9 / 31.5** | **12.5 / 28.2** | **10.9 / 23.7** | **8.9 / 19.8** | **7.6 / 16.7** | **6.7 / 14.4** |

Table A4: **Comparing robustness to sensor noise on downstream tasks in Gibson and Matterport3D.** We inject noise into the depth sensor and/or the odometer sensor. The depth-noise model consists of disparity-based quantization, high-frequency noise, and low-frequency distortion (Choi et al., 2015). The odometry noise model is based on data collected from a LoCoBot, and accumulates over time leading to a drift in pose (Ramakrishnan et al., 2020; Chaplot et al., 2020b). Note: NF denotes noise free sensing, N-D denotes noisy depth (and noise-free pose), and N-D,P denotes noisy depth and pose.

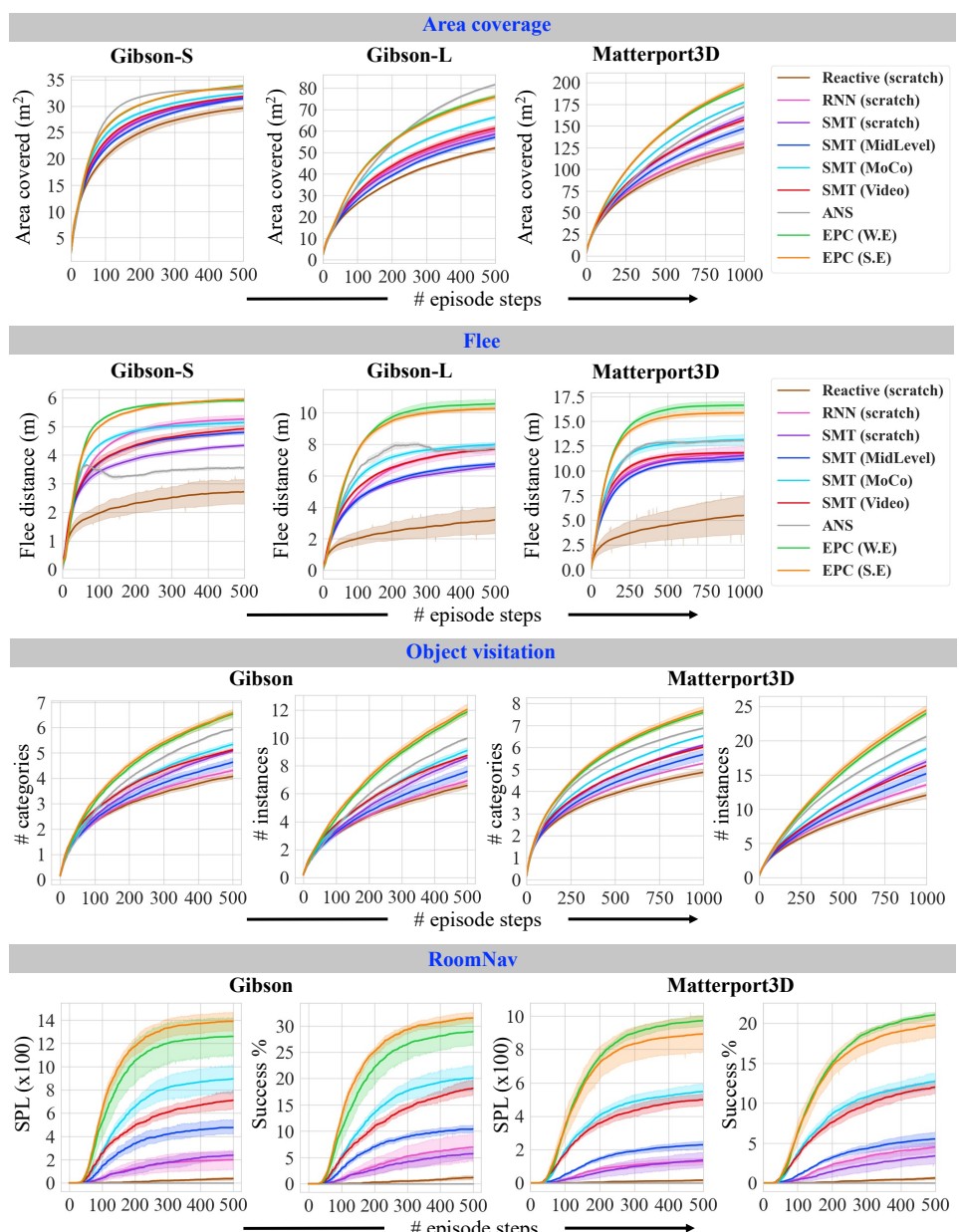

Figure A5: We highlight the downstream task performance as a function of episode time on both Matterport3D and Gibson. The legend shown near the top two rows also applies to the bottom two rows. Note that the maximum episode length on Gibson scenes is $T = 500$. For MP3D, we use a maximum episode length of $T = 1000$ for all tasks except RoomNav where we use $T = 500$ (following Narasimhan et al. (2020)).

| Method | Flee (m) MP3D | Object visitation | | RoomNav (MP3D) | |
|---|---|---|---|---|---|
| | | # categories | # instances | SPL | Success |
| EPC w/ feat-sampling | 12.4 | 7.1 | 25.9 | 9.3 | 21.3 |
| EPC | **13.9** | **7.5** | **26.6** | **10.3** | **24.4** |

Table A5: **Comparing zone feature averaging vs. sampling**. 'EPC w/ feat-sampling' is a variant of EPC which sets the target zone feature from Eqn. 2 to be the projected visual feature of a randomly frame from the zone. This leads to worse transfer results than our existing scheme which averages the projected features from all frames belonging to the target zone.

