# OpenReview forum: "Environment Predictive Coding for Visual Navigation"
_ICLR.cc/2022/Conference — ICLR 2022 Poster_

### Official Review · Reviewer_dLzX · 2021-10-30

**Correctness:** 3
**Technical Novelty And Significance:** 2
**Empirical Novelty And Significance:** 3
**Recommendation:** 8
**Confidence:** 5

**Main Review:**

Strengths

+Related work section has been clearly written. It contains not only a complete review of the literature, but also a clear discussion on why the proposed ideas are different from previous works.

+Clarity and the way the article is written are two of its strongest points. Introduction has been nicely written, positioning the work clearly, and highlighting the main contributions. The figures and diagrams are quite elaborate, and help to understand many of the ideas proposed. Sections 3 and 4 do not skimp on details. Overall: good work!

+The experimental evaluation design allows the impact of the proposed ideas to be assessed. Two different datasets are used, multiple baselines are reported, an ablation study with clear questions about possible limitations of the model, etc. Both SMT (MoCo) and SMT(Video) baselines are really informative. I appreciate when the experiments focus on trying to confirm the hypothesis proposed, and not only on beating a state-of-the-art performance.

+As for the results, they look promising. Both WE and SE strategies, combining SMT with EPC, seem to lead the SMT model to provide better results on several problems.

Weaknesses

-Paper's strongest point is not originality.
a) In this work, the scene completion problem is addressed simply using a standard transformer encoder-decoder model. An MLP is employed to obtain image-level embeddings. No significant novelty is found in the formulation of the zone decoder. All these elements are trained end-to-end following a previous contrastive loss learning approach (Gutmann & Hyvärinen, 2010). I focus on judging the novelty of the article, dissecting the EPC module, because this is the main contribution. Overall, EPC is a standard trasformer encoder-decoder trained to learn and embedding of some selected features (those belonging to the marked/masked zones).
b)The other contribution is also incremental. The encoder learned by EPC has simply been integrated into the SMT model, which is the one that performs all the navigation tasks.
I would like the authors to offer a clear argument about the novelty of their paper for a conference such as ICLR.


-The application of the EPC model to other non-SMT architectures remains unresolved. Although this is claimed in different parts of the paper, NO experiments are provided to support this claim. In my opinion, the proposed model will fit only in transformer-based solutions such as SMT. It would be useful to hear from the authors on this issue (has it been tested with any other architecture?). Perhaps some of the assertions made in the article might be relaxed.

-Something similar occurs with the type of videos used to train the model. To conjecture that any videos of building walks could be used to train the model, and not those provided by the datasets used, is also not supported by the experiments provided.

-There is another aspect for which I also find no evidence in the experiments, and which appears throughout the manuscript. Overall, the paper sells the idea that the proposed module is able to "predict the spatial structure of 3D environments". This claim is too ambitious in my humble opinion. The model is able to predict features for those masked zones. From here to understanding that the model predicts the 3D structure of the environment is a big leap. Moreover, no experiments are provided to support this claim. My advise: to revise (and relax) all those affirmations for which no experimental evidence is provided.


Minor comments:
-Figure 2 is difficult to follow, too many details in it.
-Please, punctuate all the equations. They are part of the text.

**Summary Of The Paper:**

This paper focuses on the problem of efficiently training an intelligent embodied agent for navigation purposes.
Technically, the manuscript introduces a technique named as Environment-Predictive Coding (EPC). It is a model for learning environment-level representations via self-supervision, just using video walkthroughs with masked zones.
EPC proposes to learn an environment encoder, using a transformer-based encoding-decoding, that produces environment embeddings and that predicts feature encodings for the unseen areas. This is the main scientific contribution of the paper.
This environment encoder is finally integrated in the Scene Memory Transformer (SMT) (Fang et al., 2019) for navigation purposes.
A thorough experimental evaluation is offered, using Matterport3D (MP3D) (Chang et al., 2017) and Gibson (Xia et al., 2018) for 4 navigation tasks. Reported results seem to explain the benefits of incorporating such a self-supervision mechanism.

**Summary Of The Review:**

I like the work the authors have done with the article. I don't think we have a model of great originality, but rather of an incremental nature. However, the ideas are clearly stated, and the experiments are well designed and executed. I need the authors to defend their position on my criticisms of the novelty of their ideas, and to comment on those parts for which I have not found experimental evidence. Overall: I see here a borderline submission.

---

> ### Author Response · Authors · 2021-11-23
> **Authors' response to reviewer dLzX [Part 1]**
>
> We thank the reviewer for the very encouraging comments and feedback. We provide a two-part response.
> \
> \
> **Q1. Paper’s strongest point is not originality … would like the authors to offer a clear argument about the novelty.**
>
> To clarify, EPC is **not** a standard transformer encoder-decoder model. EPC is a representation learning technique for **pre-training** a standard transformer encoder-decoder model. In fact, we share this same architecture with multiple baselines — SMT (scratch), SMT (MoCo), SMT (MidLevel), and SMT (Video) — to show the effectiveness of our pre-training method for embodied agents. In short, we do not aim to improve over the base SMT architecture.
> \
> \
> Instead, we propose EPC as a novel self-supervised learning method to learn an environment-level representation of the underlying 3D environment that an agent is navigating in. This upgrades existing embodied AI approaches that pre-train only image-level features for agents. Our key two technical contributions are on the self-supervised task side (restated from Sec. 1, last paragraph):
>
> * We propose to learn environment-level features by predicting visual features conditioned on an unseen (x, y, theta). We are the first method to do so to the best of our knowledge.
>     * We show that this spatial-conditioning is critical for SSL in Sec. 4.3 (point 1).
>
> * We learn such predictive features by proposing a simple and novel masked-zone prediction task that operates on an offline set of video walkthroughs (Sec. 3.2). We show its effectiveness over strong baselines in Sec. 4.2. We also show that this task can be robustly performed when videos are collected
>     * with noisy depth and/or odometer sensors (Sec 4.3, point 1).
>     * without odometer readings, where pose estimates have to be obtained through visual odometry (now updated in Sec. 4.3, point 1).
>     * with different types of underlying agent behavior for collecting video walkthroughs (Sec. 4.2, paragraph 3).
>
> This is akin to how researchers in image-representation learning propose different self-supervised learning strategies (MoCo, SimCLR, BYOL, etc) to pre-train standard image encoders like ResNets and ViTs. Similarly, we propose to pre-train a standard SMT policy architecture which has been shown to work well for embodied agents (Sec. 3.2, paragraph 1).
> \
> \
> **Q2. The application of the EPC model to other non-SMT architectures remains unresolved … Perhaps some of the assertions made in the article might be relaxed.**
>
> This is a possible misunderstanding.  We only suggest that our idea is “potentially applicable” to other architectures, and do not claim that it works well for all architectures (see Sec. 3.2, paragraph 1 and Sec. 2, last paragraph). We selected the SMT architecture due to its successes in visual navigation (Sec. 3.2, paragraph 1). We clarified our language about “potentially applicable” in the paper:
>
> * Related work last paragraph: We removed the line “Our approach is general and can potentially be applied to other architectures as well.”
> * Sec. 3.2, 1st paragraph: We updated the line --- “While our idea is potentially applicable to other memory models, our choice of SMT is motivated by the recent successes of transformers…”
>
> That said, our task definition is not specific to transformer-based models. For example, we can define the EPC masked-zone prediction task on a standard topological memory architecture [1, 2, 3, 4] as follows:
> \
> \
> A topological memory architecture [1, 2, 3, 4] uses a graph-based environment encoding with visual features at the nodes, and edges encoding connectivity between different observations based on the agent’s actions or relative poses. In this case, we could build a topological memory for the video walkthrough similar to [1, 2, 3, 4], then mask out visual features from specific nodes in the graph corresponding to a zone, and infer the masked features using a graph-convolutional network [5].
> \
> \
> Due to the computational cost of these experiments and the extensive nature of our analyses (as acknowledged by reviewers), we refrain from experimenting with other architectures.
> \
> \
> **Response references**
>
> [1] Savinov, Nikolay, Alexey Dosovitskiy, and Vladlen Koltun. "Semi-parametric topological memory for navigation." arXiv preprint arXiv:1803.00653 (2018).
>
> [2] Savinov, Nikolay, et al. "Episodic Curiosity through Reachability." International Conference on Learning Representations. 2018.
>
> [3] Nagarajan, Tushar, et al. "Ego-topo: Environment affordances from egocentric video." Proceedings of the IEEE/CVF Conference on Computer Vision and Pattern Recognition. 2020.
>
> [4] Chaplot, Devendra Singh, et al. "Neural topological slam for visual navigation." Proceedings of the IEEE/CVF Conference on Computer Vision and Pattern Recognition. 2020.
>
> [5] Zhang, Si, et al. "Graph convolutional networks: a comprehensive review." Computational Social Networks 6.1 (2019): 1-23.

---

> > ### Author Response · Authors · 2021-11-23
> > **Authors' response to reviewer dLzX [Part 2]**
> >
> > **Q3. To conjecture that any videos of building walks could be used to train the model … is not supported by experiments provided.**
> >
> > We acknowledge in the paper that we do not test on in-the-wild consumer videos (Sec. 3.3, paragraph 1), and we do not say that any videos of building walks could be used. We clearly discuss the requirements (or lack thereof) for the walkthroughs in Sec. 3.3, paragraph 1. To the best of our efforts, we attempt to replicate similar agent walkthroughs in a controlled manner in simulation (Sec. 3.3, paragraph 2):
> >
> > * we perform EPC pre-training on two very different types of walkthroughs (strong exploration and weak exploration), show it transfers well to 4 diverse tasks, and its performance remains similar for both types of walkthroughs (Sec. 4.2) --- akin to different types of agents gathering the videos, and how walkthrough behaviors need not be tied to downstream tasks
> >
> > * the walkthroughs are collected in environments that are not accessible for interactive training --- akin to recording walkthroughs in various inaccessible geographical regions
> >
> > * the walkthroughs are collected from a different dataset (Gibson) than that used for training the embodied agents (MP3D).  This is important because there exists a domain gap between MP3D and Gibson, and generalizing between the two is non-trivial. For example, the authors in [6] report that a SplitNet model trained only on MP3D performs much worse on Gibson (val) than a model trained on Gibson for PointNav --- akin to how the walkthrough environments can have visuals and layouts different from RL training environments
> >
> > * [new experiment] we even eliminate the need for pose information from walkthroughs by replacing the odometer sensor with predictions from an RGB-D based visual odometry model in Tab. 2 in the updated paper. We find that the results degrade slightly, but continue to outperform the baselines by a significant margin — akin to an even more challenging case where no pose information is available in these videos.
> >
> > \
> > **Q4. “predict the spatial structure of 3D environments”. This claim is too ambiguous … revise (and relax) affirmations ...**
> >
> > We appreciate the reviewer’s point. We showed examples of our model predicting features that correlate with certain 3D structures like narrow corridors, and kitchens (main Fig.4 and supp video). However, this does not imply that we are predicting 3D structures (as the reviewer points out, and we agree). We have relaxed and clarified this assertion in the paper conclusion:
> >
> > “predicting the spatial structure of 3D environments” -> “predicting features grounded in the spatial structure of 3D environments”
> >
> > In Sec. 2 (paragraph 1) and Sec. 5 (conclusion) in the updated paper, "grounded in the spatial structure" refers to how our predictions are conditioned on (x, y, theta) in the environment.
> > \
> > \
> > **Q5. Flag for Ethics Review: Yes, Privacy, security and safety**
> >
> > We have already addressed the ethical concerns in **Sec. 7: Ethics statement.**
> > \
> > \
> > **Response references**
> >
> > [6] Gordon, Daniel, et al. "Splitnet: Sim2sim and task2task transfer for embodied visual navigation." Proceedings of the IEEE/CVF International Conference on Computer Vision. 2019.

---

> > > ### Comment · Reviewer_dLzX · 2021-11-24
> > > **Issues raised in my original review have been adequately addressed**
> > >
> > > First of all, I would like to thank the authors for their detailed answers.
> > > Regarding originality, the defense proposed by the authors is convincing. Actually, whether an article offers a **marginal** contribution or not is difficult to argue. What is certain here is that the self-supervised learning model proposed, applied to navigation purposes, is somehow novel. The idea of learning to infer some features for unseen zones (masked zones in this paper) is not novel. It is true that there is a lot of previous literature on models that are able to do video prediction (i.e. a future video frame is an unseen zone at the end of the day, like in this paper) and some of these works are used for navigation purposes. We can take as an example the recent Pathdreamer model [MR1]. In any case, I'm happy with the response provided. Maybe, some more relevance references about video prediction could be added to the "Learning representations for visual navigation" paragraph in section 2.
> > >
> > >
> > > Regarding the suggested relaxations in the applicability of the model, I also agree with the proposed modifications ("potentially applicable"). It is important that the experiments support all claims/conclusions. In the same direction, it also clear what happens with the requirements for the walkthroughs videos needed. The discussion in the paper is adequate.
> > >
> > > Overall, all the issues raised in my original review have been adequately addressed. I would be willing to increase the rating of this work.
> > >
> > >
> > > [MR1] Pathdreamer: A World Model for Indoor Navigation Jing Yu Koh, Honglak Lee, Yinfei Yang, Jason Baldridge, Peter Anderson. ICCV 2021.

---

> > > > ### Author Response · Authors · 2021-11-27
> > > > **Authors' follow-up**
> > > >
> > > > Thank you for acknowledging our responses and increasing the rating. We will cite the suggested works in the final version of the paper.

---

### Official Review · Reviewer_JRbw · 2021-11-02

**Correctness:** 4
**Technical Novelty And Significance:** 3
**Empirical Novelty And Significance:** 3
**Recommendation:** 6
**Confidence:** 2

**Main Review:**

Please find below a detailed review of the strengths and weaknesses of the paper.

**Strengths**

- The paper addresses a challenging and pertinent problem i.e. how to learn representations of large environments that can help high level tasks. I think the approach is well-designed. I strongly agree with the hypothesis that pose information should help with learning representations grounded in 3D space, and the paper does a good job at verifying this hypothesis.

- The experimental evaluation is really well done in almost all aspects. The authors have picked very good datasets to evaluate on that are challenging and representative of real-world applications. The tasks they have picked clearly demonstrate the advantages of the epc method and have practical interest. The methods against which the proposed approach are compared are appropriate. The ablation study is also excellent. I think the questions that the authors raise and answer in the ablation study are very relevant and helps to show why the specific components of the method (such as the spatial conditioning) are necessary. Overall, the experimental evaluation is well thought out and demonstrates the advantages of the method.

- The results are very promising. The pretraining using EPC seems to help across all the tasks tested on and the improvement is statistically significant.


**Weaknesses**

- The main weakness of the paper is that there are lots of works in computer vision on “place recognition” (see [A]). In general, these methods also groups areas into “zones” and learn representative features of each zone. How would the proposed EPC features compare to those learned by a place recognition method such as [B]?  Also, a very related method is that of [C]. It would be good to include a discussion on this approach and how it compares.
-  It would be good to show how the zone segmentation affects the performance of the method. Does it matter where the zone boundaries are chosen?

[A] Lowry, Stephanie, et al. "Visual place recognition: A survey." IEEE Transactions on Robotics 32.1 (2015)

[B] Chen, Zetao, et al. "Deep learning features at scale for visual place recognition." 2017 IEEE International Conference on Robotics and Automation (ICRA)  2017.

[C] Lenton, Daniel, et al. "End-to-End Egospheric Spatial Memory." International Conference on Learning Representations (ICLR) 2021.



**Summary Of The Paper:**

The main contribution of this paper is a method for learning a spatial representation of an environment, called environment predictive encoding, that can be used for pre-training a network in an unsupervised manner for downstream embodied tasks such as navigation. The results show that the method significantly improves the sample efficiency of four common downstream tasks.

**Summary Of The Review:**

Overall I think this is an excellent paper that presents an idea that is both simple, practical and effective. The method is well validated on two challenging benchmark environments, MP3D and Gibson, and compared against numerous state of the art approaches and achieves great performance. In the response, i would like the authors to address the points in “weaknesses” above.

---

> ### Author Response · Authors · 2021-11-23
> **Authors' response to Reviewer JRbw**
>
> We thank the reviewer for the encouraging comments and suggested references.
> \
> \
> **Q1. The main weakness … lots of works in computer vision on “place recognition”. Compare proposed EPC features to these.**
>
> Thank you for the references.
> \
> The references aim to perform place recognition, whereas we aim to learn environment encoders for embodied agent navigation.  However, we can relate the two areas of work and are happy to cite those references.  Zones can be seen as a specific type of “place” in 3D environments.  We use a simple definition of a place (appeared in the agent’s view b/w time t1 to t2), and infer features of places that the agent may not have seen before, conditioned on the agent’s visual history and camera poses. In contrast, prior work [A, B] learns features to recognize the same place under different conditions. We have added a discussion in Appendix A13 due to space constraints in the main paper.
> \
> [C](ICLR 2021) is a memory model that could be used instead of SMT. We have added this to Related Work, citing it along with the other memory architectures in the last paragraph of Sec. 2.
> \
> \
> **Q2. It would be good to show how the zone segmentation affects the performance of the method. Does it matter where the zone boundaries are chosen?**
>
> In response Q2 to R-q87t, we discussed an alternative zone segmentation technique which uses 3D point-cloud overlap between pairs of frames for clustering them into zones. This enforces stricter constraints of the overlap between zones. We initially tried such an approach for EPC, but found it to work worse while incurring significant complexity in our implementation (more details in Appendix A8). It also requires access to the camera intrinsics for each video, which we do not require in our current implementation. In short, our current implementation is simpler and more effective.
> \
> \
> Additionally, we have updated appendix A5 in the paper with the hyperparameter search we used to determine the zone sizes (N, M) for our current strategy. To quickly summarize the results, we find that having a smaller zone size benefits area coverage, flee, and object visitation, whereas RoomNav benefits from having larger zone sizes (40). Please see Appendix A5 for the complete results.
> \
> \
> **Response references**
> \
> \
> [A] Lowry, Stephanie, et al. "Visual place recognition: A survey." IEEE Transactions on Robotics 32.1 (2015)
>
> [B] Chen, Zetao, et al. "Deep learning features at scale for visual place recognition." 2017 IEEE International Conference on Robotics and Automation (ICRA) 2017.
>
> [C] Lenton, Daniel, et al. "End-to-End Egospheric Spatial Memory." International Conference on Learning Representations (ICLR) 2021.

---

### Official Review · Reviewer_q87t · 2021-11-02

**Correctness:** 3
**Technical Novelty And Significance:** 2
**Empirical Novelty And Significance:** 3
**Recommendation:** 8
**Confidence:** 5

**Main Review:**

**Strengths**

***S1***: Pre-training aims at exploiting the higher level semantic & geometric features of an environment by predicting a scene representation \hat(f)_i^u that should not have been observed before, such that lower, image-level interpolation of frames (without reasoning of semantics) is prevented. This is an interesting concept, which is applicable to many reasoning tasks.

***S2***: Good experimental evaluation and ablation study including the SMT architecture with different pre-training techniques. This comparison demonstrates the usefulness of the proposed “environment-level” pretraining.

***S3***: SoTA results on the chosen datasets for various robotic visual navigation tasks.


**Weaknesses**

***W1***: Insufficient argumentation and analysis for the proposed zone based masking approach.

It is argued that a (random) frame based masking “can result in poor representation learning since shared content from nearby unmasked frames can make the prediction task trivial”. A simple spatial sampling of frames would ensure that the visual overlap is minimized. The zone decoder would then become a frame decoder which predicts features for a masked frame, conditioned on the pose of the masked frame. This would still qualify as learning an “environment-level” representation as blending/inpainting is not possible. A comparison to such an approach is missing. The argument that zones result in a higher-level representation is not verified. Therefore, the advantage of using zones over frames in the pre-training is not at all clear.

***W2***: The definition of zones solely on the number of frames does not account for the motion of the camera and thus does not avoid overlapping of zones.

E.g. if the camera stands still for >N frames, then consecutive zones capture the same scene part and zone feature prediction based on the pose seems trivial. A definition of zones based on the camera pose (and not necessarily consecutive frames) would be more convincing, as it aims to minimize scene overlap.

The paper does not make any analysis of different zone selection strategies and the amount of overlap between seen and unseen zones is unknown. Though, this is important, as the authors argue that their zone based masking is superior to a frame based masking because of the reduced overlap (see W1).

***W3***:  What prevents the zone decoder from ignoring the environment embeddings Epsilon and predicting \hat(f)_i^u as only an embedding/projection of p_i^u?
Given that the supervising f_i^u is built from the RGBD frames AND their poses, see Eq. (2), this seems to be the trivial solution; i.e. the MLP would ignore the visual content X and just rely on the poses. The paper would benefit from explaining the choice of supervision better and in particular point out why the features f are a good choice for supervision.

***W4***: The idea behind the usage of an egocentric view, i.e. usage of relative over absolute poses, is unclear.

In pre-training the poses of frames of seen zones are relative to the mean pose p_i^u of the unseen zone. When plugging the pre-trained encoder in the SMT architecture poses relative to the query o_t can still be used during training (unclear if this is actually done). Though, during inference o_t changes, which would require to update the relative poses of all previous observations and re-encode them into “environment embeddings”. This seems largely impractical. In contrast, one would only want to encode an observation once which argues for a scene-centric coordinate representation, e.g. use the first frame’s pose as reference.

***W5***: Experimental evaluation on the very same dataset as the baseline SMT work is not provided.

**Comments**

***C1***: In Fig. 3 left it is illustrated that all frames of a zone are fed to a MLP, while it should be per frame to obtain the image-level embeddings.



**Summary Of The Paper:**

The work builds upon the scene memory transformer (SMT, Fang et al., 2019) for visual navigation, which encodes and adds past observations to a memory and uses attention to exploit spatio-temporal dependencies.

The authors propose a self-supervised pre-training for the observation encoder using an encoder-decoder architecture with the encoded memory as the bottleneck (the decoder is only used for pre-training). Self supervision is achieved by using sequences of observations (posed RGBD images) and splitting them into batches of temporally consecutive frames (termed “zones” in the paper). Considering a subset of the zones as not observed, i.e. not available as input for the network, the goal of the encoder-decoder is to predict the features of an unseen zone from the seen input zones, conditioned on the pose of the unseen zone.

Compared to a frame based masking, the zone (=spatio-temporal consecutive frames) based exclusion of data results in less overlap between seen and unseen scene parts, which is argued to learn higher-level semantic and geometric representation of the 3D environment

Contributions:
- Approach for self supervision based on the prediction of visual-geometric features for unseen scenes.
- Superior performance to SoTA on the navigation task.


**Summary Of The Review:**

The paper presents an interesting idea for self-supervised pre-training of the encoder stage of the SMT architecture. The conceptual approach is not limited to visual navigation but applicable to a variety of (3D) reasoning tasks.

The paper lacks clarity in the details of the approach, see above weaknesses W1-W4. With that I’m concerned that the proposed masked-zone prediction task (which is the core contribution of the paper) is a good choice for handling the pre-training. While the approach achieves SoTA on the visual navigation task there are only limited learnings but numerous uncertainties for the reader why this is so.

I would be willing to increase my rating, if the above weaknesses are addressed and it is shown that the proposed masked-zone pre-training is superior to alternative encoder pre-training approaches..

---

> ### Author Response · Authors · 2021-11-23
> **Authors' response to Reviewer q87t**
>
> We thank the reviewer for the insightful feedback and questions.
> \
> \
> **Q1. Insufficient argumentation and analysis for the proposed zone based masking approach (W1).**
>
> We are not sure what the reviewer means by “simple spatial sampling of frames” in the context of a video walkthrough. One possible interpretation could be this: Cluster images based on camera viewpoints (i.e., 3D spatial coverage), and pick a single frame from each cluster. Then mask one of these frames and predict it from the rest. We agree that this does count as environment-level representation learning. However, this **would not** count as “frame-based masking” since the frames are jointly clustered first into “spatial zones” (one frame is not masked independent of the rest). It’s not just that a single frame feature is being predicted, but more importantly, the features from an entire zone are removed. The above can be seen as a variant of our approach, where a zone is defined based on spatial information rather than time.  In fact, we attempted this variant when developing EPC and found it did not perform quite as well as the version presented in the paper (see below).  We opted to pick a simple and sensible strategy that works well in practice.
> \
> \
> **Q2. Definition of zones … does not account for motion of the camera … does not avoid overlapping of zones (W2).**
>
> Thank you for raising this. The reviewer’s suggestion is definitely a reasonable approach to creating zones. In fact, this was the initial version we tried, but it did not work as well as our much simpler current method. Our initial approach worked as follows: We measured the 3D point-cloud overlap between pairs of frames (projected using depth, camera pose, and camera intrinsics), and clustered the frames that have significant overlap together using hierarchical agglomerative clustering. We defined each cluster to be a zone, and used the same SSL training pipeline. Here are the results on the MP3D validation split on 3 tasks compared to our original method.
>
>
> | Method | Flee (m) | # categories | # instances | RoomNav SPL | RoomNav success |
> |---|---|---|---|---|---|
> | EPC (W.E) w/ spatial | 12.0 | 7.4 | 26.4 | 9.8 | 23.7 |
> | EPC (W.E) w/ temporal | 14.4 | 7.5 | 26.3 | 11.0 | 25.0 |
> | EPC (S.E) w/ spatial | 12.5 | 7.3 | 25.7 | 9.3 | 22.5 |
> | EPC (S.E) w/ temporal | 13.9 | 7.5 | 26.6 | 10.3 | 24.4 |
>
> \
> The suggested variant “EPC w/ spatial” performs reasonably well. However, it is significantly more complex than our proposed model (requiring camera intrinsics, 3D point-cloud overlap computations and clustering), and performs worse than our simpler time-based zone creation approach “EPC w/ temporal”. Since the videos are unconstrained, the agent might revisit the same areas multiple times, causing too many frames to be grouped together into a single zone. This results in very irregular zone sizes for masked-zone prediction across videos, which affects the quality of EPC SSL training. By not strictly limiting the overlap between zones, our proposed method learns by both inferring parts of the environment that were never seen, as well as those that were seen from different viewpoints. Our current method is simpler, and works better. We have added this analysis to Appendix A8, and alluded to it in Sec. 3.1 in the updated paper.
> \
> \
> **Q3. What prevents the zone decoder from ignoring the environment embeddings … trivial solution … just rely on the poses … (W3)**
>
> Thank you for pointing this out. It is a bug in our notation in the paper.  In our implementation, we do set the poses to zero while computing the target feature in Eqn. 2. We have corrected this in the updated paper.
> \
> \
> **Q4. The idea behind the usage relative over absolute poses is unclear (W4).**
>
> This is an implementation detail from Sec. 3.4 in the SMT paper. Quoting the text from the SMT paper:
> “A special caution is to be taken with the pose vector. First, at every time step all pose vectors in the memory are transformed to be in the coordinate system deﬁned by the current agent pose. This is consistent with an ego-centric representation of the memory. Thus, the pose observations need to be re-embedded at every time step, while all the other observations are embedded once.”
> The SMT authors indicate that this is an important choice. This is also efficient since the RGB-D image features for an observation are extracted only once, while only the pose embeddings are updated (using an FC layer) at each step. The reviewer’s suggestion is also reasonable, but this is orthogonal to our idea since we do not aim to improve upon the SMT architecture.
> \
> \
> **Q5. Experimental evaluation on the very same dataset as SMT is not provided (W5).**
>
> The SUNCG dataset used in the SMT work is not available. https://lawstreetmedia.com/news/tech/intellectual-property/facebook-and-princeton-sued-over-allegedly-scraped-data/
> \
> \
> **Q6. Fig. 3 comment**
>
> Thank you for spotting this. We have fixed it.

---

> > ### Author Response · Authors · 2021-11-23
> > **Authors' response summary**
> >
> > **Summary response - “I would be willing to increase my rating, if the above weaknesses are addressed and it is shown that the proposed masked-zone pre-training is superior to alternative encoder pre-training approaches.”**
> >
> > We believe we have addressed the weaknesses raised by the reviewer. To recap our responses,  \
> > \
> > **W1:** `frame-based’ masking that the reviewer suggested is actually a variant of EPC which uses spatial-zones.  \
> > \
> > **W2:** We already tried the suggested spatial-zone variant. Our current EPC zone definition is simpler and superior to the spatial-zone variant. \
> > \
> > **W3:** We fixed the bug in our paper notation. Our implementation avoids the trivial solution suggested by the reviewer. \
> > \
> > **W4:** This is an implementation detail from SMT, and we reiterate the reasoning behind this choice and why it is practical to implement. \
> > \
> > **W5:** The dataset used in SMT is not available anymore.

---

> > ### Comment · Reviewer_q87t · 2021-11-24
> > **Follow up questions on rebuttal.**
> >
> > First of all I'd like to thank the authors for their thorough rebuttal. I appreciate the additional work they have put into their manuscript to increase clarity. Most of my concerns are resolved. There's one important point about the choice of coordinate frames which still confuses me. If this can be clarified I'll increase my rating.
> >
> > **Q1**: Your assumption is correct, with "*spatial* sampling of frames" I was referring to a sampling of frames such that the visual overlap of seen and unseen frames is limited. This can be achieved by defining zones, and as such is similar to the proposed zone-based masking (which can be defined by temporal closeness or covisibility).
> > The difference is in Eq (2) where the target feature f_i^u would be the single frame feature that is being predicted, rather than an average over all features from a zone. Also p_i^u would be the pose of that single frame, not the average over the zone. I'm missing a comparison to this as I can't follow why one needs the averaging in Eq (2). Could you provide an argumentation and/or experimental results for this?
> >
> > **Q2**: Thank you for adding the additional experiment and results. It demonstrates that you had considered this approach and found it to be inferior to the proposed solution. This resolves my concerns.
> >
> > **Q3**: Thanks for the clarification in Eq. (2). However, I'm still concerned that a trivial solution is possible here: The model works with an egocentric view of the world, that is all poses are relative to p_i^u. The egocentric view also implies for me that p_i^u is the identity transformation.
> >
> >  * (A) If this is the case, then it's trivial for the model to predict f_i^u which is built from poses set to zero.
> >  * (B) If p_i^u is not identity, but the pose of the unseen zone in the world coordinate frame, then I do not understand what information this absolute pose provides to the model. In particular as p_i^u is used for attention, but the embeddings e_i live (or originate from) as different coordinate frame.
> >
> > I might be missing an important point here -- as might be the case for other readers of your work. Thus I'd like you to ask to clarify the used coordinate frames and explain how a trivial solution is avoided. Thanks.
> >
> > **Q4**: Thanks for the clarification. I agree that a different coordinate frame representation compared to the SMT work is beyond the scope of this paper. My confusion also stems from the points raised in Q3. Concerns from W4 are resolved.
> >
> > **Q5**: Thanks. Didn't know. Resolved.

---

> > > ### Author Response · Authors · 2021-11-27
> > > **Authors' response to follow-up questions**
> > >
> > > We thank the reviewer for appreciating our responses. Please find our responses to the follow-up questions below.
> > >
> > > > Q1: Your assumption is correct, with "spatial sampling of frames" I was referring to a sampling of frames such that the visual overlap of seen and unseen frames is limited ...  I'm missing a comparison to this as I can't follow why one needs the averaging in Eq (2). Could you provide an argumentation and/or experimental results for this?
> > >
> > > This was a hyperparameter choice for us. The target feature could be a single frame from the target zone, or it could be the averaged features of the target zone (which is our  proposed method) --- both are valid approaches, and we did try them. As noted in Footnote 1 from page 4, we found averaging the features from the target zone was comparable or better than randomly sampling features within a zone. That is why we selected feature averaging over random sampling. For example, on Flee with the weak exploration videos, we found that predicting features for a randomly sampled frame from the target zone gave us 12.5m flee distance on MP3D val set, whereas predicting the average features gave us 14.4m flee distance. We will add the complete comparison in the appendix for camera ready.
> > >
> > > > Q3: Thanks for the clarification in Eq. (2). However, I'm still concerned that a trivial solution is possible here: The model works with an egocentric view of the world, that is all poses are relative to p_i^u. The egocentric view also implies for me that p_i^u is the identity transformation.
> > >
> > > You are right in saying that the egocentric view results in the identify pose for p_i^u. The reason why this trivial solution does not happen is because of the presence of negatives in Eq (4). When the features are computed for any zone (positive or negative), the visual features are present and the poses are set to zero (as shown in the updated Eq (2)).
> > >
> > > For the sake of argument, let us assume that the MLP in Eq (2) was learning to set the target feature to some constant vector (since pose is always 0 and it ignores the visual features). In that case, the target features for both positive and negative zones will be the same constant vector. The loss in Eq (4) would be $L_i = -\textrm{log}\frac{1}{1 + N},$ regardless of the zone decoder output ($N$ is the total # negatives). In order to reduce the loss further, the MLP *must* learn to encode the visual features in the target zones. We do observe this happening in our experiments. For example, when we have 24 negatives and 1 positive, the loss value in the trivial case will be $3.21$. However, in our training, the loss value goes down to $0.18$.

---

> > > > ### Comment · Reviewer_q87t · 2021-11-29
> > > > **Re: Follow up questions,**
> > > >
> > > > I'd like to thank the authors for their effort they put into resolving my concerns. The additional comparison between feature averaging vs. random sampling will increase the reproducibility of the work for future readers.
> > > > My concern about the trivial solution is resolved as well. I did overlook the fact that only by encoding the visual appearance one can minimize the contrastive loss.
> > > >
> > > > My final request for the final version of this paper is to make the implications of the egocentric view easiest grasp, especially that p_i^u is the identity transformation.
> > > >
> > > > My questions are resolved and I'm positive about the paper now.

---

### Official Review · Reviewer_MWw9 · 2021-11-04

**Correctness:** 3
**Technical Novelty And Significance:** 3
**Empirical Novelty And Significance:** 3
**Recommendation:** 8
**Confidence:** 4

**Main Review:**

Strengths:
- This paper is a resubmission from last years’ ICLR, from what I can see many of the concerns from previous reviews have been addressed, and the paper has been thoroughly revised. To highlight:
  - Many new baselines are added, SMT(Moco) is an interesting baseline showing that just with image-based representation learning it doesn’t perform so well. ANS is a very competitive baseline for exploration tasks.
  - More tasks are added, especially the semantic navigation tasks.
  - Added different exploration strategies
- With the added experiments and baselines, the experiments and analysis are very thorough
- The paper is generally well written and easy to follow.
- The ablation study is very informative, it shows that using spatial conditioning is important for SSL for embodied tasks. It can provide guideline for the general recipe for pretraining embodied agents.

Weaknesses:

- “We learn the representations on a collection of video walkthroughs” is somewhat misleading, since the video walkthrough contains poses and people would think of video walkthroughs as those “house touring videos”. The authors said later “We now realize this process in photorealistic Gibson scenes; we leave leveraging in-the-wild consumer videos as a challenge for future work.”
- It would be interesting to use the poses from visual odometry, apart from adding noise in ablation study 2. This will further lift the assumptions on walkthrough videos with pose
- The authors should probably adapt ANS to make it work with the stop action, instead of letting it trivially fail.
- (minor) In figure 5 different baselines seem to have different steps. Also the SMT (MoCo) doesn't seem to be converged yet.

**Summary Of The Paper:**

The authors propose a self-supervised representation learning method called environment predictive coding inspired by the context prediction in other representation learning works. This explicit spatial conditioning encourages learning representations that capture the geometric and semantic regularities of 3D environments. The learned representations can be used for downstream navigation tasks, achieving higher sample efficiency over standard image-representation learning.


**Summary Of The Review:**

This is a good paper with thorough experimental analysis. It is significantly improved from the previous version and I would recommend accepting it.

---

> ### Author Response · Authors · 2021-11-23
> **Authors' response to reviewer MWw9**
>
> We thank the reviewer for the encouraging and valuable feedback.
> \
> \
> \
> **Q1. “We learn the representations on a collection of video walkthroughs” is somewhat misleading.**
>
> We have updated the introduction to clarify what video walkthroughs are: “The videos contain RGB-D and odometry.” While these videos are not in-the-wild house touring videos, we do our best to replicate the scenario of robotic agents performing tasks in real-world environments (please see response Q3 to R-dLzX). RGB-D and odometers are standard on-board sensors.
> \
> \
> **Q2. It would be interesting to use the poses from visual odometry … will lift assumptions on walkthrough videos with pose.**
>
> Thank you for the suggestion. We replaced the GT pose from the videos with poses estimated from a visual odometry model using RGB-D readings [6]. We have updated Table 2 in the main paper to include this experiment (EPC w/ noisy-d+vo). The results remain similar or improve relative to using a noisy odometer.
> \
> \
> **Q3. Should adapt ANS to make it work with the stop action.**
>
> We tried our best to get ANS to succeed for RoomNav by trying different hyperparameter choices. Prior work which extended ANS for ObjectNav [5] used a dedicated semantic mapping model for deciding when to STOP.  To get similar functionality for RoomNav, we need to build accurate room maps and use them to decide when to STOP. To the best of our knowledge, this is non-trivial to implement and no existing public implementations are available.
> \
> \
> **Q4. Figure 5 - SMT (MoCo) doesn’t seem to be converged yet.**
>
> We do not train models to convergence, and restrict our training budget to 13M-15M frames to demonstrate the advantage of pre-training representations (Sec. 4.1). The goal is to learn faster and be sample efficient. This training budget for downstream tasks is at least as large as recent work on transfer learning (10M frames in [1,2], 0.5M frames in [3]), meaning we are giving models adequate training time by current standards.
> \
> \
> **Q5. Figure 5 - baselines seem to have different steps.**
>
> The reason for slightly different end points is due to the pre-emptive nature of DD-PPO training, where slow workers (“stragglers”) can be forced to collect fewer steps per update [4]. Thus, the number of steps for training can vary slightly depending on the FPS for different workers (even for the same number of updates). For example, SMT (MoCo) trains for slightly more steps on area coverage, but slightly fewer steps for object visitation and flee. Nevertheless, the results are comparable since we use the same amount of resources for each method (60 parallel environments, 2000 updates), and train on similar machines throughout.
> \
> \
> **Response references**
>
> [1] Du, Yilun, Chuang Gan, and Phillip Isola. "Curious Representation Learning for Embodied Intelligence." arXiv preprint arXiv:2105.01060 (2021).
>
> [2] Sax, Alexander, et al. "Learning to Navigate Using Mid-Level Visual Priors." Conference on Robot Learning. PMLR, 2020.
>
> [3] Nagarajan, Tushar, and Kristen Grauman. "Learning Affordance Landscapes for Interaction Exploration in 3D Environments." Advances in Neural Information Processing Systems 33 (2020): 2005-2015.
>
> [4] Wijmans, Erik, et al. "DD-PPO: Learning Near-Perfect PointGoal Navigators from 2.5 Billion Frames." International Conference on Learning Representations. 2019.
>
> [5] Chaplot, Devendra Singh, et al. "Object goal navigation using goal-oriented semantic exploration." Advances in Neural Information Processing Systems 33 (2020).
>
> [6] Zhao, Xiaoming, et al. "The Surprising Effectiveness of Visual Odometry Techniques for Embodied PointGoal Navigation." Proceedings of the IEEE/CVF International Conference on Computer Vision. 2021.

---

> > ### Comment · Reviewer_MWw9 · 2021-11-24
> > **Thanks for the update**
> >
> > Thanks for the clarification and I appreciated the added experiments and results, my review remains unchanged and positive.

---

### Author Response · Authors · 2021-11-23
**Overall response**

We thank the reviewers for their valuable feedback and insightful suggestions. Overall, the reviewers felt positively about the paper and appreciated several features of the work:

* Experiments are thorough/very thorough/good/significant improvements/really well done (R-MWw9, R-q87t, R-JRbw, R-dLzX)
* Paper clarity (R-MWw9, R-dLzX)
* Ablations are (very) informative/good/excellent (R-MWw9, R-q87t, R-JRbw, R-dLzX)
* Interesting idea applicable to many tasks (R-q87t)
* SoTA results for various robotic visual navigation tasks (R-q87t)
* Addresses a challenging and pertinent problem (R-JRbw)
* Related work clearly written (R-dLzX)
* Excellent paper (R-JRbw)
* Like the work the authors have done (R-dLzX)

We respond to each reviewer individually, and update the paper to incorporate the feedback. The changes in the paper made in response to the reviewers’ comments are highlighted in red. We make limited adjustments to parts of the paper such as re-wording, updating the figures, and moving implementation details to the appendix. This is done to fit the 9-page limit with very limited changes to the actual content.

---

### Decision · Program_Chairs · 2022-01-20

**Decision:**

Accept (Poster)

**Comment:**

The authors have done a good job methodologically addressing reviewer concerns. The empirical results are good, and the application impact is clear. There were some concerns about the technical heft of the approach, but there's overall agreement that the effective application to the domain is interesting and done very well. The AC is a bit concerned about the impact of the regularities of the domain used on the results, especially with regard to semantic regularities (homes have very particular regularities). But even without answering this question (it should be discussed in the camera ready though), this paper makes a solid contribution.